 **eLIFE**

# BRAF activates PAX3 to control muscle precursor cell migration during forelimb muscle development

Jaeyoung Shin[†], Shuichi Watanabe[†], Soraya Hoelper, Marcus Krüger, Sawa Kostin, Jochen Pöling, Thomas Kubin*, Thomas Braun*

Max-Planck-Institute for Heart and Lung Research, Bad Nauheim, Germany

**Abstract** Migration of skeletal muscle precursor cells is a key step during limb muscle development and depends on the activity of PAX3 and MET. Here, we demonstrate that BRAF serves a crucial function in formation of limb skeletal muscles during mouse embryogenesis downstream of MET and acts as a potent inducer of myoblast cell migration. We found that a fraction of BRAF accumulates in the nucleus after activation and endosomal transport to a perinuclear position. Mass spectrometry based screening for potential interaction partners revealed that BRAF interacts and phosphorylates PAX3. Mutation of BRAF dependent phosphorylation sites in PAX3 impaired the ability of PAX3 to promote migration of C2C12 myoblasts indicating that BRAF directly activates PAX3. Since PAX3 stimulates transcription of the *Met* gene we propose that MET signaling via BRAF fuels a positive feedback loop, which maintains high levels of PAX3 and MET activity required for limb muscle precursor cell migration.

*For correspondence: thomas. kubin@mpi-bn.mpg.de (TK); thomas.braun@mpi-bn.mpg.de (TB)

[†]These authors contributed equally to this work

Competing interests: The authors declare that no competing interests exist.

## Introduction

In vertebrates, skeletal muscles of the trunk and limbs originate from condensations of the paraxial mesoderm, the somites (*Braun and Gautel, 2011*; *Buckingham and Relaix, 2007*). Epaxial muscles are derived from the dorso-medial part of the somatic dermomyotome while the ventro-lateral part gives rise to hypaxial muscles (*Ordahl and Le Douarin, 1992*). Hypaxial muscles of the body wall and intercostal muscles are generated by elongation of the dermomyotomal epithelium. In contrast, muscles of the limbs, the diaphragm and the tongue are generated from a population of long-range migrating muscle precursor cells, which delaminate from the ventral dermomyotome at specific positions along the cranial-caudal axis after epithelial-mesenchymal transition, allowing them to form muscles far away from somites (*Chevallier et al., 1977*; *Christ and Brand-Saberi, 2002*; *Christ et al., 1983*).

Several genes including *Pax3* (*Bober et al., 1994*), *Met* (*Bladt et al., 1995*; *Dietrich et al., 1999*), *Cxcr4* (*Vasyutina et al., 2005*), *Gab1* (*Sachs et al., 2000*), *Six1;Eya1* (*Heanue et al., 1999*) and *Lbx1* (*Brohmann et al., 2000*; *Gross et al., 2000*; *Schäfer and Braun, 1999*) have been identified to control somite maturation and compartmentalization, delamination of muscle precursor cells from the dermomyotomal epithelium as well as muscle precursor cell migration, proliferation and differentiation. More specifically, *Pax3* is required for correct formation of the ventro-lateral dermomyotome (*Bober et al., 1994*; *Daston et al., 1996*) as well as for survival (*Relaix et al., 2005*) and migration of limb muscle precursor cells (*Daston et al., 1996*). *Met* is necessary for de-epithelialization and migration of limb muscle precursor cells (*Bladt et al., 1995*) but also for myocyte fusion (*Webster and Fan, 2013*). It is also known that PAX3 controls expression of *Met* in the ventro-lateral dermomyotome (*Relaix et al., 2005*; *Yang et al., 1996*) by direct binding to the *Met* gene promoter (*Epstein et al., 1996*), thereby enabling delamination and migration of limb muscle precursor cells.

However, the full complexity of the interactions within the genetic network orchestrating limb muscle precursor cell migration and the functional regulation of the activity of PAX3 and its multiple iso-forms (*Wang et al., 2006*) has not been uncovered yet.

MET signaling is highly complex and involves several scaffolding adaptors and surface signal modifiers, which allows MET to activate multiple different biochemical pathways including the MAPK (ERK, JNK and p38 MAPKs) pathway, the PI3K-AKT axis, the STAT pathway and the IkB-NFkB complex (reviewed in (*Birchmeier et al., 2003*; *Trusolino et al., 2010*)). Importantly, mutants of MET unable to bind the adaptor GRB2, which is considered to act as the primary mediator of RAS-RAF activation, does not affect migration of limb muscle precursor cells but inhibits proliferation of fetal myoblasts and formation of secondary myofibers (*Maina et al., 1996*). In contrast, inactivation of the adaptor *Gab1* severely impairs migration of limb muscle precursor cells (*Sachs et al., 2000*). GAB1 acts as a docking platform for several molecules including PI3K, PLC, CRK, and SHP2 but also activates the RAS-RAF route after activation by the tyrosine phosphatase SHP2 (*Birchmeier et al., 2003*; *Trusolino et al., 2010*). This raises several questions: Does the RAS-RAF pathway contribute to migration of limb muscle precursor cells? If RAF is involved in regulation of limb muscle precursor cell migration, which of the three serine/threonine kinases (ARAF, BRAF, CRAF) does the job? Are potential effects of RAF transmitted via the canonical MEK-ERK pathway or by different means?

To answer these questions we inactivated the *Braf* gene specifically in limb muscle precursor cells, since germ line inactivation of *Braf* results in embryonic lethality between E10.5 and E12.5 and causes multiple defects including growth retardation, vascular and neuronal defects (*Wojnowski et al., 1997*). We found that *Braf* is required for muscle precursor cell migration and skeletal muscle formation in the forelimbs. Protein-protein interaction studies revealed that BRAF phosphorylates and activates PAX3 after endosomal trafficking to a perinuclear position and translo-cation into the nucleus. Our results suggest a positive feedback loop, which drives skeletal muscle formation by maintaining high levels of PAX3 and MET activity in migrating limb muscle precursor cells.

## Results

### BRAF mediates growth factor induced muscle precursor cell migration in vitro

The tyrosine kinase receptor MET is instrumental for delamination of limb muscle precursor cells from the dermomyotome and subsequent migration. To identify the branches of the MET signaling network driving migration of myogenic cells, we turned to the muscle cell line C2C12, since evaluation of signaling processes in migrating limb muscle precursor cells is difficult due to the small size of the cell population and its transient appearance. We found that HGF, the ligand of the MET receptor, robustly induced migration of C2C12 cells. Stimulation of migration was blocked by knock-down of *Met* demonstrating that C2C12 cells can be utilized to study the mechanisms of MET recep-tor signaling for migration of myogenic cells (*Figure 1A*). Systematic analysis of the role of potential downstream effectors of MET signaling by siRNA-mediated knockdown disclosed an important role of the serine/threonine-specific protein kinase BRAF, which essentially phenocopied the effects of *Met* receptor knockdown (*Figure 1A*). Interestingly, knockdown of *Pax3* did also inhibit HGF-medi-ated stimulation of C2C12 cell migration suggesting a functional involvement of PAX3 in MET signal-ing (*Figure 1A*). To test whether expression of BRAF alone is sufficient to promote migration we transfected WT *Braf* and the constitutively active (CA) *Braf* (V600E) mutation into C2C12 cells. We observed a strong stimulation of migration by CA BRAF (V600E) and -to a lesser extent- WT BRAF while expression of CRAF had only minor effects on muscle cell migration (*Figure 1A*). CA BRAF (V600E) and WT BRAF also stimulated expression of *Pax3* and *Met* in C2C12 cells adding further evi-dence to the putative role of BRAF in the regulatory loop controlling migration of myogenic cells (*Figure 1A*). Next, we investigated whether BRAF and PAX3 also promote migration of primary der-momyotomal cells (*Mennerich et al., 1998*). We found that RCAS virus-mediated expression of BRAF and PAX3 but not human alkaline phosphatase (AP) increased cellular migration out of somitic explants isolated from chicken embryos and cultivated in matrigel further corroborating the results obtained with C2C12 cells (*Figure 1—figure supplement 1*).

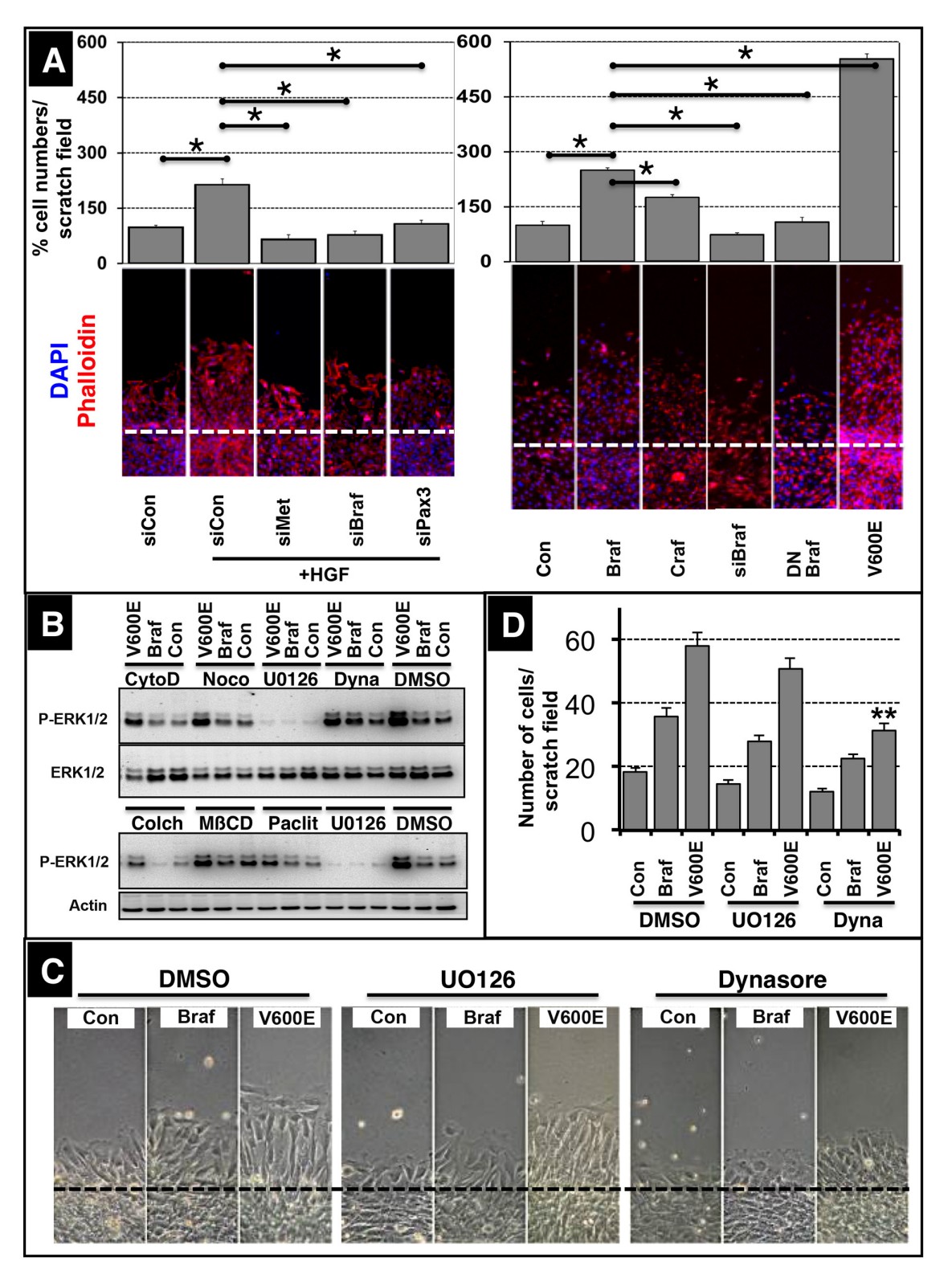

**Figure 1.** BRAF mediates muscle precursor cell migration independent of MEK/ERK signaling. (**A**) Immunofluorescence staining of migrating muscle cells (C2C12) after knock down of the hepatocyte growth factor (HGF) receptor (siMet), *Braf* (siBraf) and *Pax3* (siPax3) or after transfection of cultures with Craf, Braf, a dominant negative form of Braf (DN Braf) and CA Braf (V600E) in the presence or absence of HGF. Con indicates control vector and siCon represents a scrambled siRNA control. Cultures were analyzed 4 hr after scratching excluding effects of cell proliferation. Cell numbers were

*Figure 1 continued on next page*

*Figure 1 continued*

determined by counting the number of DAPI-stained nuclei. A statistical assessment is shown in the upper part of the panel (n = 15; Mann-Whitney-U test, p*<0.05). siRNA knock-down efficiencies for *Met* (80%), *Braf* (70%) and *Pax3* (85%) were determined by Western blot analysis. (B) Phosphorylation of ERK1/2 after transfection of C2C12 cells with control vector (Con), WT Braf and CA Braf (V600E) in the presence or absence of UO126 (5 µM) or cytochalasin D (5 µM; CytoD), noco (5 µM; nocodazole), Dynasore (80 µM; dyna), colchicine (0.01%; colch), methyl-β-cyclodextrin (3 mM; MβCD), and paclitaxel (5 µg/ml; paclit). Addition of DMSO served as an additional control. (n = 2). Cytochalasin D disrupts actin filaments. Nocodazole, colchicine, and paclitaxel interfere with microtubuli assembly or disassembly. U0126 inhibits the MEK/ERK pathway. Dynasore blocks dynamin-dependent endocytosis. Methyl-β-cyclodextrin removes cholesterol from cultured cells and disrupts lipid rafts. (C) Statistical assessment of the experiments shown in (D) (n = 15; Mann-Whitney-U test, p**<0.01). (D) Microscopic imaging of migrating C2C12 cells after transfection with a control vector (Con), WT Braf and CA Braf (V600E) in the presence or absence of DMSO, UO126 and Dynasore. Cultures were analyzed 4 hr after scratching excluding effects of cell proliferation.

The following figure supplement is available for figure 1:

**Figure supplement 1.** BRAF and PAX3 stimulate limb muscle precursor cell migration.

Since BRAF activates the MEK-ERK signaling cascade in muscle cells (*Figure 1B*), we wanted to know whether inhibition of MEK1/2 blocks the effects of BRAF on migration. Surprisingly, addition of the MEK1/2 inhibitor UO126 had only minor effects on migration, although the increase in ERK1/2 phosphorylation after transfection of WT and CA BRAF (V600E) was efficiently prevented by the inhibitor (*Figure 1C,D*) indicating a MEK/ERK independent mechanism of BRAF signaling. Since MET undergoes rapid endocytosis in a process called 'endosomal signaling' and traffics through peripheral endosomes to a perinuclear localization (*Barrow-McGee and Kermorgant, 2014*), we investigated whether inhibition of endosomal trafficking abrogates the effects of BRAF on migration of C2C12 muscle cells. Interestingly, addition of Dynasore, a pharmacological inhibitor of endosomal trafficking, significantly prevented BRAF dependent stimulation of migration when the constitutively active form of BRAF was used in the experiments (*Figure 1C,D*).

## BRAF is required for migration of limb muscle precursor cells and formation of forelimb muscles

Although C2C12 cells represent a useful model to study mechanistic aspects of muscle cell migration, the inhibition of HGF-stimulated migration after knockdown of *Braf* does not prove that *Braf* is also instrumental to regulate limb muscle formation in the embryo in vivo. We therefore took advantage of a mouse strain (Braf[nfl]) in which three *loxP*-sites had been inserted to flank exon three as well as a neomycin selection cassette (*Figure 2A*) (*Pfeiffer et al., 2013*). Breeding with MeuCre mice (*Leneuve et al., 2003*) yielded Braf[fl] mice, in which the neomycin cassette was removed but exon three is flanked by *loxP*-sites, and Braf[del] mice lacking exon 3, which encodes parts of the Ras-binding domain (*Figure 2A*). Deletion of exon three created a functional null allele of *Braf* leading to embryonic lethality of Braf[del/del] mice consistent with previous reports (*Pfeiffer et al., 2013*; *Wojnowski et al., 1997*). Braf[fl] mice were crossed to the Pax3-Cre strain allowing specific inactivation of *Braf* in the dermomyotome and limb muscle precursor cells (*Figure 2B*). Immunofluorescence staining using an anti-BRAF antibody confirmed the absence of BRAF protein in the dermomyotome of Pax3-Cre//Braf[del/del] mutant embryos at E10.5 while the dermomyotome of WT control embryos was strongly positive for BRAF (*Figure 3—figure supplement 1*). Importantly, germ line inactivation (Braf[del/del]) as well as deletion of *Braf* in *Pax3*-expressing limb muscle precursor cells (Pax3-Cre//Braf[del/del]) resulted in arrest of cell migration as indicated by the absence of PAX3[+]-cells in the developing forelimb buds at E10.5 (*Figure 3A,B*), which was further validated by immunofluorescence analysis of PAX3[+] cells (*Figure 3—figure supplement 1*) and RT-PCR analysis of *Met* and *Pax3* expression (*Figure 2C*). Closer inspection of the PAX3 immunofluorescence staining revealed that only very few *Braf*-deficient PAX3[+]-cells delaminated from the dermomyotome and initiated migration suggesting a requirement of *Braf* for epithelia-mesenchymal transition of dermomyotomal cells and/or migration (*Figure 3—figure supplement 1*). Staining for *Lbx1* expression, which is also expressed in limb muscle precursor cells and needed for targeted migration, confirmed this conclusion (*Figure 3C*). We also found a virtually complete absence of *Myod* and *Myf5* expressing myogenic cells in limb buds of E10.5 Pax3-Cre//Braf[del/del] embryos (*Figure 3C*), although we noted a

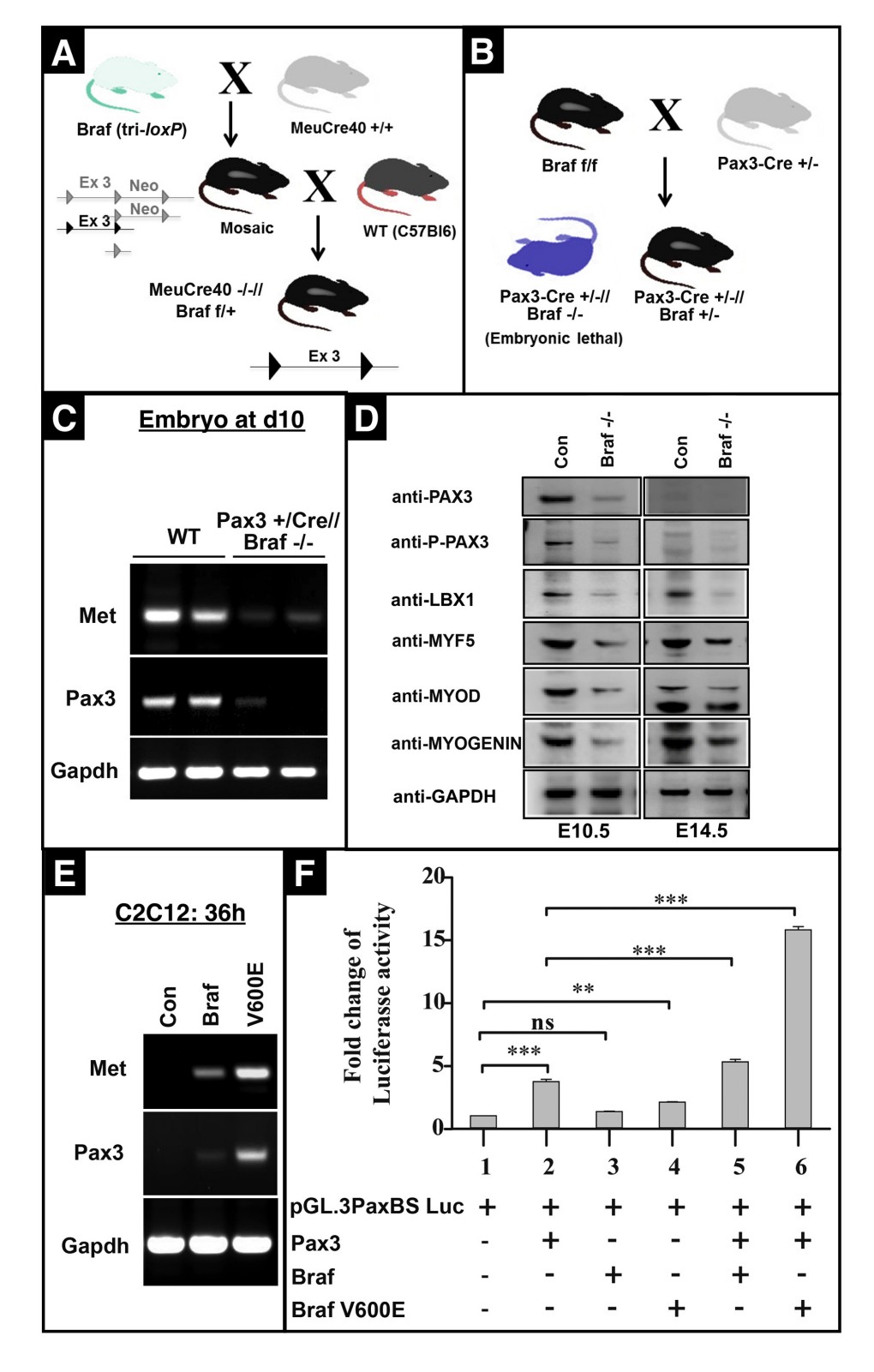

**Figure 2.** Inactivation of *Braf* in limb muscle precursor cells. (**A**) Strategy for generation of *Braf* floxed mice (Braf[nfl]). Breeding with MeuCre mice yielded Braf[fl] mice, in which the neomycin cassette was removed but exon three is flanked by *loxP*-sites. (**B**) Braf[fl] mice were bred with Pax3-Cre mice to generate animals lacking *Braf* in *Pax3* expressing cells. Pax3-Cre//Braf[del/del] embryos die around E15.5. Mutant embryos were analyzed between E10.5 and E14.5. (**C**) RT-PCR analysis of *Met* and *Pax3* expression in limb buds of Pax3-Cre//Braf[del/del] embryos at E10.5 n = 2. (**D**) Western blot analysis of
*Figure 2 continued on next page*

Figure 2 continued

expression of different markers in limb buds of Pax3-Cre//Braf^del/del embryos at E10.5 and E14.5 n = 2. (E) Expression of WT and CA Braf (VE600E) in C2C12 cells increases expression of *Met* and *Pax3*. n = 3. (F) BRAF enhances PAX3-dependent transcriptional responses. The pGl.3 Pax3BS luc reporter construct containing two PAX3 binding sites in front of a minimal promoter was co-transfected with different combinations of Pax3, Braf, and CA Braf (V600E) expression vectors into HEK293T cells. The activity of Firefly Luciferase was normalized by a co-transfected renilla luciferase in all experiments. Data represent the mean ± SEM and analyzed using ANOVA with a Tukey-Kramer post-hoc comparison test. ***p<0.001.

reduced expression at later stages by Western blot analysis (*Figure 2D*) indicating that a small population of *Braf*-deficient limb muscle progenitor cells was able to reach its target. Accordingly, we observed a major reduction of forelimb muscles at later stages of development (E14.5), which was more pronounced in the distal compared to the proximal parts of the limb, where some residual muscle formation was seen (*Figure 3D,E*). In contrast, we did not observe a significant increase in the number of apoptotic cells in *Braf* mutant compared to WT control embryos at E10.5 suggesting that the loss of *Braf* in dermomyotomal cells does not lead to programmed cell death (*Figure 3—figure supplement 1*). Furthermore, we investigated the presence of endothelial precursor cells in Pax3-Cre//Braf^del/del mutant embryos at E10.5, since transplantation of *Kdr*-mutant mouse presomitic mesoderm into chicken embryos had revealed a crucial role of endothelial cells for muscle progenitor cell migration (*Yvernogeau et al., 2012*). KDR primarily signals via the PKC-MAPK and the PI3K pathway (*Vieira et al., 2010*), although some evidence exists that KDR can also activate MAPK via CRAF and MEK in some cells types (*Takahashi et al., 1999*), which raised the possibility that inactivation of *Braf* might indirectly disrupt muscle progenitor cell migration by preventing migration of endothelial precursor cells downstream of KDR. However, similar to the presence of normal numbers of endothelial progenitor cells in *Pax3* mutant embryos (*Yvernogeau et al., 2012*), we did not observe a reduction of CD31-positive endothelial cells in *Braf* mutant limb buds (*Figure 3—figure supplement 1*) indicating that KDR does not critically rely on BRAF for promoting endothelial precursor cell migration.

To further explore effects of BRAF on critical components of the regulatory network driving migration of myogenic cells we transfected WT Braf and CA Braf (V600E) into C2C12 myoblasts. RT-PCR analysis revealed a strong up-regulation of *Met* and *Pax3* expression indicating that BRAF induced transcription of both genes, most likely by activation of key transcription factors (*Figure 2E*). In addition, we analyzed whether BRAF might directly increase the transcriptional activity of PAX3. We therefore constructed a luciferase reporter construct containing two PAX3 consensus binding sites in front of a minimal promoter. Co-transfection of the Pax3 reporter plasmid together with Pax3 and Braf or CA Braf (V600E) into HEK293T cells revealed a strong increase of transcriptional activity, resulting in an up to 15-fold change when CA Braf (V600E) was used (*Figure 2F*). Taken together our data indicate that BRAF is essential for limb muscle formation presumably by mediating MET signaling via PAX3 to enable migration of limb muscle precursor cells.

## BRAF physically interacts with PAX3 in muscle precursor cells in vivo and in vitro

Since our data indicated that the effects of BRAF on migration of myogenic cells are not mediated via the canonical MEK-ERK signaling pathway, we decided to search for additional targets. Importantly, mass spectrometry analysis of immunoprecipitated samples from WT limb buds of E10.5 mouse embryos using an antibody against endogenous BRAF identified -among several other proteins- the nuclear transcription factors PAX3 and PAX7 as potential interaction partners of BRAF (*Figure 4A*, *Figure 4—figure supplement 1*). Similar observations were made when BRAF was precipitated from C2C12 cells transfected with the CA Braf (V600E). Reciprocal mass spectrometry experiments, in which we incubated protein extracts from WT limb buds of E10.5 mouse embryos with GST-PAX3 protein followed by immunoprecipitation with an anti-PAX3 antibody and immunoprecipitation of PAX3 from extracts of C2C12 cells transfected with HA-tagged Pax3 yielded identical results and also uncovered interactions of PAX3 and BRAF with components of the endocytosis pathway such as ARF5, six and DNM2 along with several other proteins (*Figure 4A*, *Figure 4—figure supplement 1*). Coupled immunoprecipitation-Western blot analysis of samples from E10.5 limb buds using either BRAF antibodies for IP and PAX3 antibodies for Western blot or vice versa further

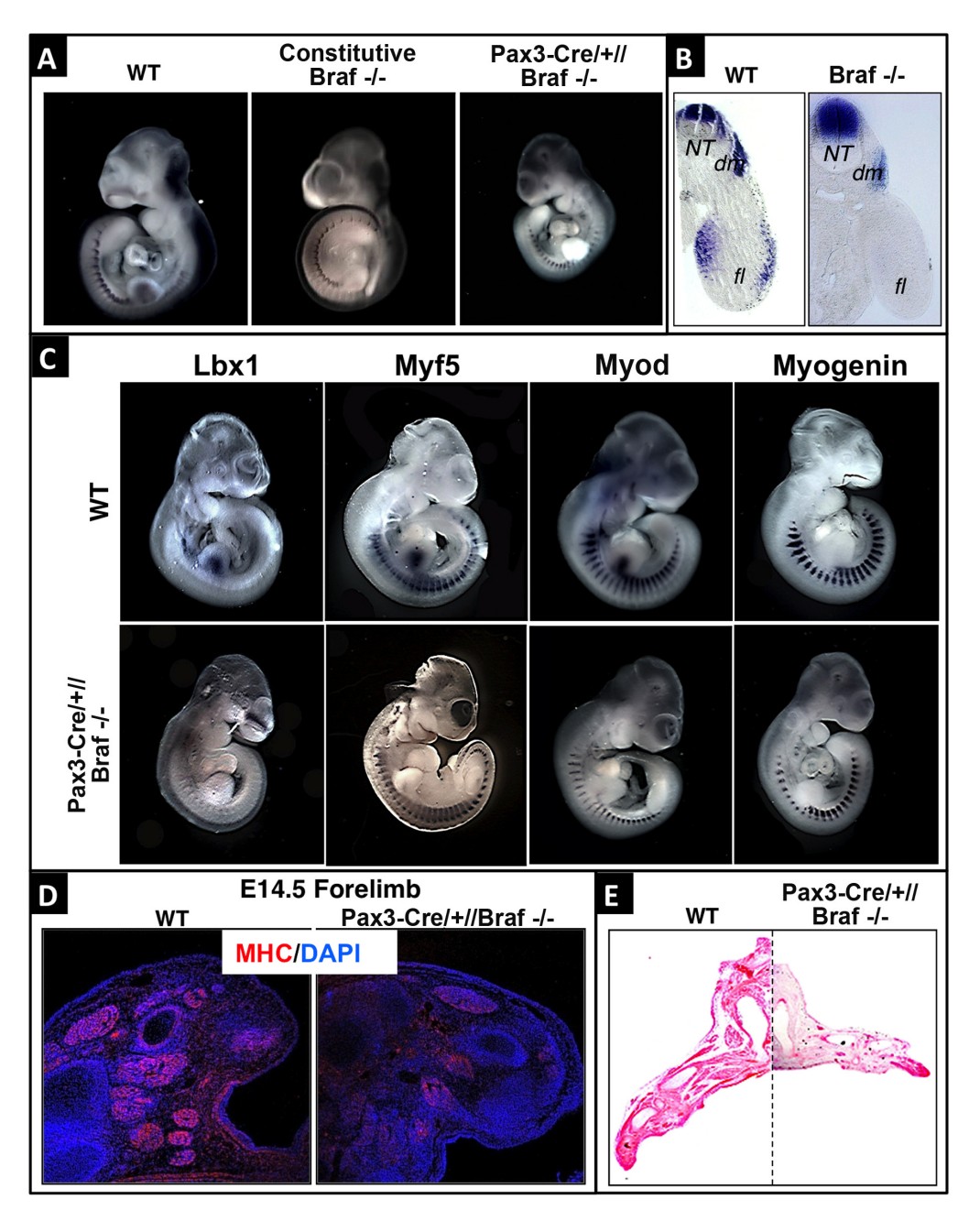

**Figure 3.** BRAF is required for limb muscle precursor cell migration during mouse embryogenesis. (**A**) Pax3 whole mount in situ hybridization of WT, germ line Braf[del/del] and Pax3-Cre//Braf[del/del] mutant embryos at E10.5. (**B**) Transverse sections of WT and Pax3-Cre//Braf[del/del] mutant embryos at E10.5 after Pax3 whole mount in situ hybridization. Inactivation of *Braf* results in loss of PAX3[+] cells in forelimbs. NT: neural tube; dm: dermomyotome; fl: forelimb. (**C**) Whole mount in situ hybridization of E10.5 WT and Pax3-Cre//Braf[del/del] mutant embryos using Lbx1, Myf5, Myod, and Myogenin probes. (**D**) Immunofluorescence staining of forelimbs from WT and Pax3-Cre//Braf[del/del] mutants for myosin heavy chain (MHC) at E14.5. (**E**) Hematoxylin and eosin staining of forelimbs from WT and Pax3-Cre//Braf[del/del] mutants at E14.5.

The following figure supplement is available for figure 3:

**Figure supplement 1.** Inactivation of *Braf* in *Pax3* expressing cells impairs limb muscle precursor cell but not endothelial cell migration during mouse embryogenesis.

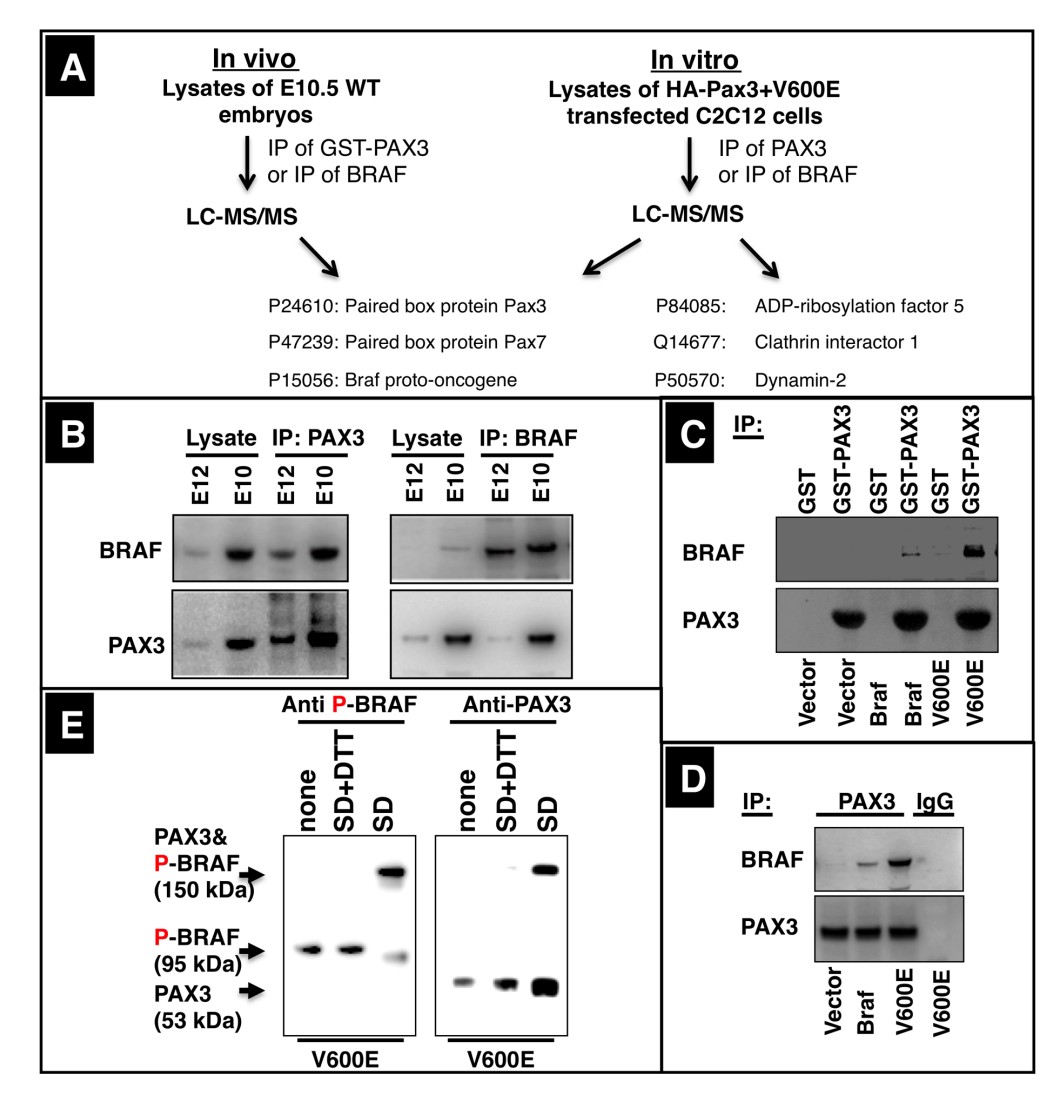

**Figure 4.** BRAF directly interacts with PAX3 in migrating muscle precursor cells. (**A**) PAX3, GST-PAX3 and PAX3 were immunoprecipitated from protein extracts of E10.5 wild type embryos or from C2C12 muscle cells after transfection with HA-Pax3 or CA Braf (V600E). Immunoprecipitations were analyzed by mass spectrometry after SDS-PAGE and in gel digestions. A selected list of proteins identified by Mascot search analysis is presented. (**B**) Analysis of the interaction of PAX3 and BRAF in E10.5 and E12.5 WT embryos by coupled immunoprecipitation/Western blot analysis. n = 3. (**C**) Western blot analysis of GST and GST-PAX3 immunoprecipitations after transfection of C2C12 cells with empty vector, Braf and CA Braf (V600E). n = 3. (**D**) Western blot analysis of PAX3 or BRAF immunoprecipitations from C2C12 transfected with CA Braf (V600E) after chemical cross-linking (SD) and cleavage of the cross-linker with DTT (SD+DTT). None = no cross-linker added. n = 3. (**E**) Western blot analysis of PAX3 immunoprecipitations from C2C12 transfected with WT Braf, CA Braf (V600E) and vector control (vector). n = 3.

The following figure supplement is available for figure 4:

**Figure supplement 1.** Selected lists of BRAF interaction partners identified by mass spectrometry analysis.

confirmed the physical interaction between BRAF and 53 kDa PAX3 (*Figure 4B*). Next, we wanted to know whether activation of BRAF leads to more efficient binding to PAX3. Comparative GST pull-down assays of extracts from C2C12 cells transfected either with WT Braf or CA Braf (V600E) revealed a significantly stronger interaction of the constitutively active compared to the WT form of BRAF with GST-PAX3 (*Figure 4C*). Identical results were obtained when endogenous PAX3 was immunoprecipitated from C2C12 cells after transfection with WT Braf or CA Braf (V600E)

(*Figure 4E*). Finally, we performed cross-linking experiments with NHS-diazirine cross-linker (SD) in C2C12 cells transfected with CA Braf (V600E) followed by immunoprecipitation with BRAF or PAX3 antibodies. Western blot analysis using corresponding antibodies detected the 95 kDa BRAF and the 53 kDa PAX3 proteins as separate bands (*Figure 4D*). Moreover, either antibody detected an additional single band at ca. 150 kDa, which corresponds to the cross-linked BRAF-PAX3 heteromer. Cleavage of the spacer with DTT severed the cross-linked BRAF-PAX3 complex, further verifying identity of the band (*Figure 4D*).

## A fraction of activated BRAF accumulates in the nucleus after endosomal transport

Although our co-immunoprecipiation experiments established a physical interaction between PAX3 and BRAF, the assumed location of BRAF in the cytoplasm seems to prevent a meaningful physiological association with PAX3, which is located in the nucleus. Hence, we analyzed specifically whether a fraction of BRAF is transported into the nucleus. Subcellular fractionation of C2C12 cells disclosed the presence of a minor amount of endogenous BRAF in the nucleus although the major share of BRAF was present in the cytoplasm (*Figure 5A*). In contrast, PAX3 was exclusively found in the nuclear fraction, even after expression of large amounts of HA-tagged Pax3. The concentration of BRAF in the nuclear fraction increased when CA Braf (V600E) was transfected into cells irrespective of the presence or absence of HGF. Interestingly, the nuclear level of BRAF decreased when endosomal trafficking was blocked by administration of Dynasore (*Figure 5A*). We also conducted additional co-immunoprecipitation experiments using only the nuclear fraction of transfected C2C12 cells, which corroborated the interaction of nuclear BRAF with PAX3 in the absence of cytosolic BRAF (*Figure 5B*). The co-immunoprecipitation experiments also revealed enhanced interaction of CA BRAF (V600E) with PAX3, which might be due to increased transport of CA BRAF (V600E) into the nucleus. Finally, we performed immunofluorescence staining for PAX3 and BRAF using mouse embryonic tissues or C2C12 cells transfected with Braf, CA Braf (V600E) and Pax3. We observed a striking co-localization of PAX3 and BRAF in nuclei of limb muscle precursor cells and strong signals for P-BRAF in nuclei of transfected cells although localization of P-BRAF in the cytoplasm dominated (*Figure 5C,D*, *Figure 5—figure supplement 1*). Co-staining for P-BRAF and the endosomal marker EEA1 revealed a close association of endosomal vesicle and P-BRAF in the cytoplasm but not in the nucleus (*Figure 5—figure supplement 1*).

Intrigued by the fact that inhibition of endosomal trafficking by Dynasore abrogated the effects of BRAF on migration of myogenic cells and prevented accumulation of BRAF in the nucleus we targeted the early endosome antigen 1 (EEA1), an essential component of the endosomal pathway in C2C12 cells using siRNAs. Knockdown of *Eea1* resulted in decreased levels of nuclear BRAF while siErk1/2 showed little effects (*Figure 6A*). Endosomal trafficking of MET has been described to be required for full activation of signals such as GAB1, ERK1/2, STAT3 and RAC1 (*Barrow-McGee and Kermorgant, 2014*). To explore whether disruption of endosomal trafficking does not only inhibit transport of BRAF into the nucleus but also activation of ERK1/2 we knocked down *Eea1*, *dynamin-2* (*Dnm2*), *clathrin heavy chain* (*Cltc*), *caveolin-1* (*Cav1*) and *ADP ribosylation factor 6* (*Arf6*). In neither case we observed significant effects on the phosphorylation level of ERK1/2 while knock down of *Braf* or *Erk1/2* resulted in a strong reduction of ERK1/2 phosphorylation (*Figure 6B*). Furthermore, knockdown of *Eea1* but not of *Erk1/2* significantly inhibited BRAF-mediated stimulation of C2C12 cell migration (*Figure 6C*) supporting our conclusion that signaling events downstream of MET necessary for limb muscle precursor migration rely on the translocation of BRAF into the nucleus and not on the activation of the ERK1/2 signaling cascade.

## BRAF regulates muscle precursor cell migration through PAX3 phosphorylation

The interaction of the BRAF kinase with PAX3 and the inhibition of HGF-mediated migration of myogenic cells by knockdown of *Pax3* suggested that BRAF might exert its effects downstream of MET by phosphorylation and subsequent activation of PAX3. To characterize potential phosphorylation sites within the PAX3 protein, we isolated PAX3 by immunoprecipitation from C2C12 cells, HEK293T cells and from mouse embryonic limb buds at E10.5. Subsequent fractionation of FASP-digested PAX3 peptides by cation-exchange liquid chromatography followed by mass spectrometry analysis

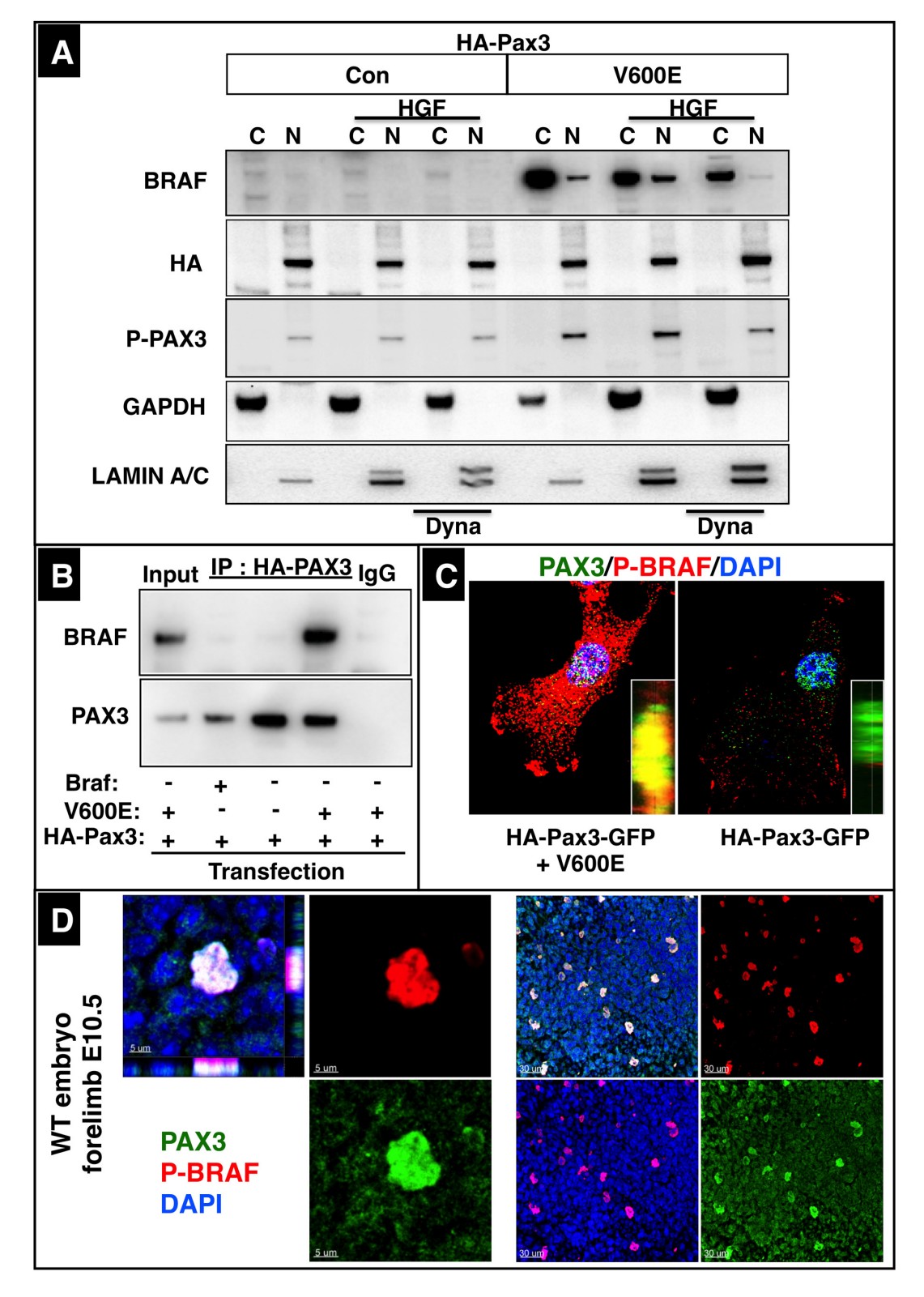

**Figure 5.** A fraction of BRAF co-localizes with PAX3 in nuclei of muscle cells. (**A**) Western blot analysis of cytoplasmic (C) and nuclear fractions (N) of C2C12 cells transfected with CA Braf (V600E), WT Braf, HA-Pax3 or HA-Pax3-GFP. n = 3. Successful fractionation was monitored by cytoplasmic GAPDH and the nuclear protein LAMIN A/C. Some cultures were treated with Dynasore (Dyna) for 30 min before fractionation as indicated. (**B**) Western blot analysis of immunoprecipitations of nuclear fractions isolated from migrating C2C12 cells after transfection with WT Braf, CA Braf (V600E), or HA-Pax3.
*Figure 5 continued on next page*

*Figure 5 continued*

n = 3. (C) High resolution confocal images of C2C12 cells transfected with WT, CA Braf (V600E) and HA-Pax3-GFP. (D) High resolution confocal image of a PAX3 and BRAF positive forelimb muscle precursor cell at E10.5 (left panel). A lower magnification is shown in the right panel.

The following figure supplement is available for figure 5:

**Figure supplement 1.** Intracellular localization of BRAF in C2C12 cells.

using a quadrupole-based Q Exactive instrument identified multiple new phosphorylation sites at serine and threonine residues as well as five phosphorylation sites (at Ser180, Ser187, Ser201,

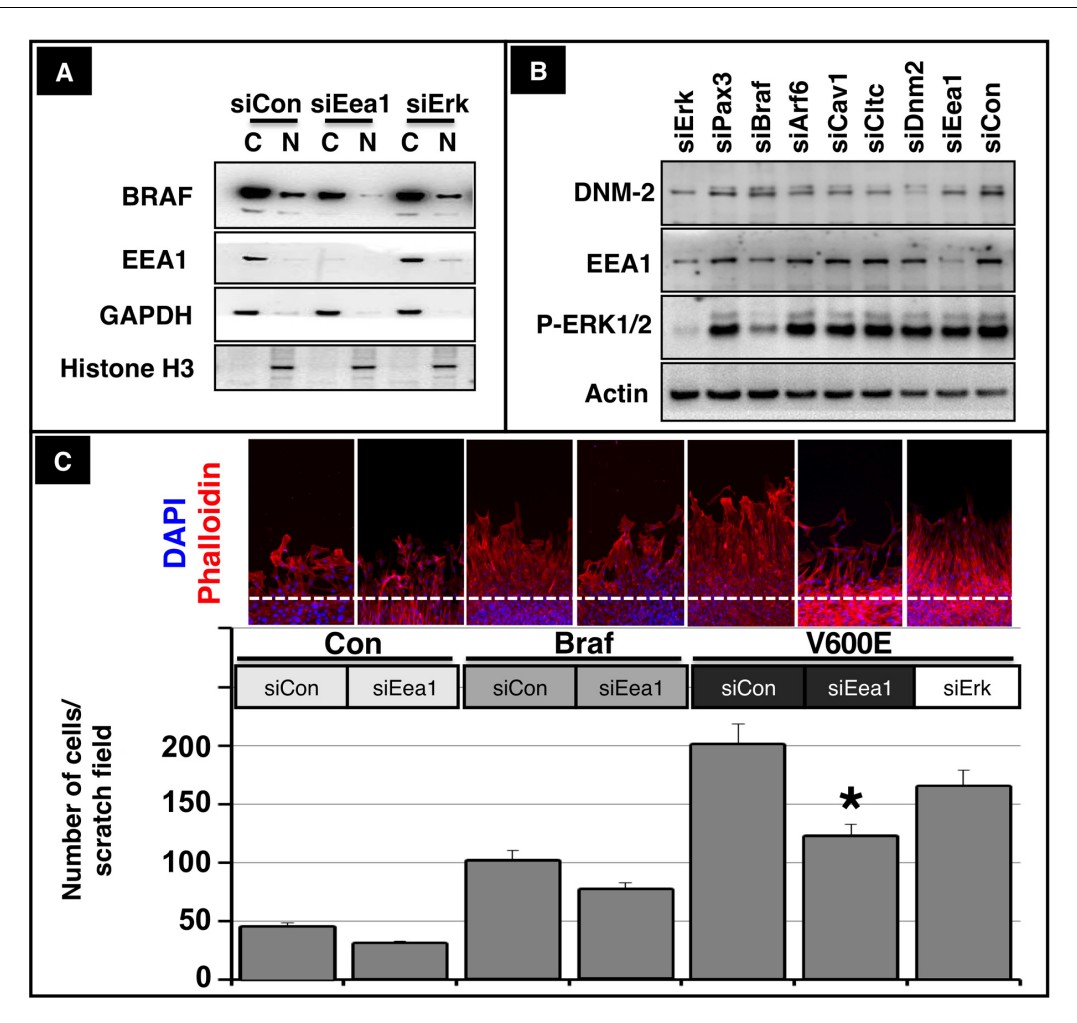

**Figure 6.** Nuclear translocation of BRAF and migration of muscle cells depend on intact endosomal trafficking. (A) Western blot analysis of isolated subcellular fractions of C2C12 cells after transfection of CA Braf V600E and knockdown of *Eea1* or *Erk1/2*. n = 3. Knockdown of *Eea1* prevented accumulation of BRAF in the nucleus (N). C: cytoplasm. siRNA knock-down efficiencies for *Eea1* (60%) and *Erk* (95%) were determined by Western blot analysis. (B) Knock down of *Braf* (siBRAF) but not of *Eea1* (siEea1), *Dnm-2* (siDnm2), *Cltc* (siCltc), *Cav1* (siCav1) and *Arf6* (siArf6) did prevent phosphorylation of ERK1/2. Western blot analyses of siRNA transfected C2C12 cells are shown. n = 3. Actin served as loading control. siRNA knock-down efficiencies for *Pax3* (85%), *Braf* (70%), *Arf6* (80%), *Cav1* (75%), *Cltc* (70%), *Dnm2* (75%) and *Eea1* (60%) were determined by Western blot analysis. (C) Immunofluorescence staining of migrating C2C12 cells after knockdown of *Eea1* or *Erk-1/2*. Cultures were transfected with control vector, Braf or CA Braf (V600E) as indicated. Con indicates control cultures and siCon means control siRNA. siRNA knock-down efficiencies for *Eea1* (60%) and *Erk* (95%) were determined by Western blot analysis. Cell numbers were determined by counting the number of DAPI-stained nuclei. A statistical analysis of siEea1 versus siCon in CA Braf (VE600E) transfected cultures is shown. n = 12; Mann-Whitney-U test, (p*<0.05).

Ser205 and Ser209), which had already been reported before (*Figure 7A*). The majority of the phosphorylation events corresponded well to sites predicted by NetPhos 2.0 (www.cbs.dtu.dk/services/NetPhos), PHOSIDA (www.phosida.de) and PhosphoVariant (http://phosphovariant.ngri.go.kr). To understand whether enhanced BRAF activity increases phosphorylation of PAX3, we quantified the

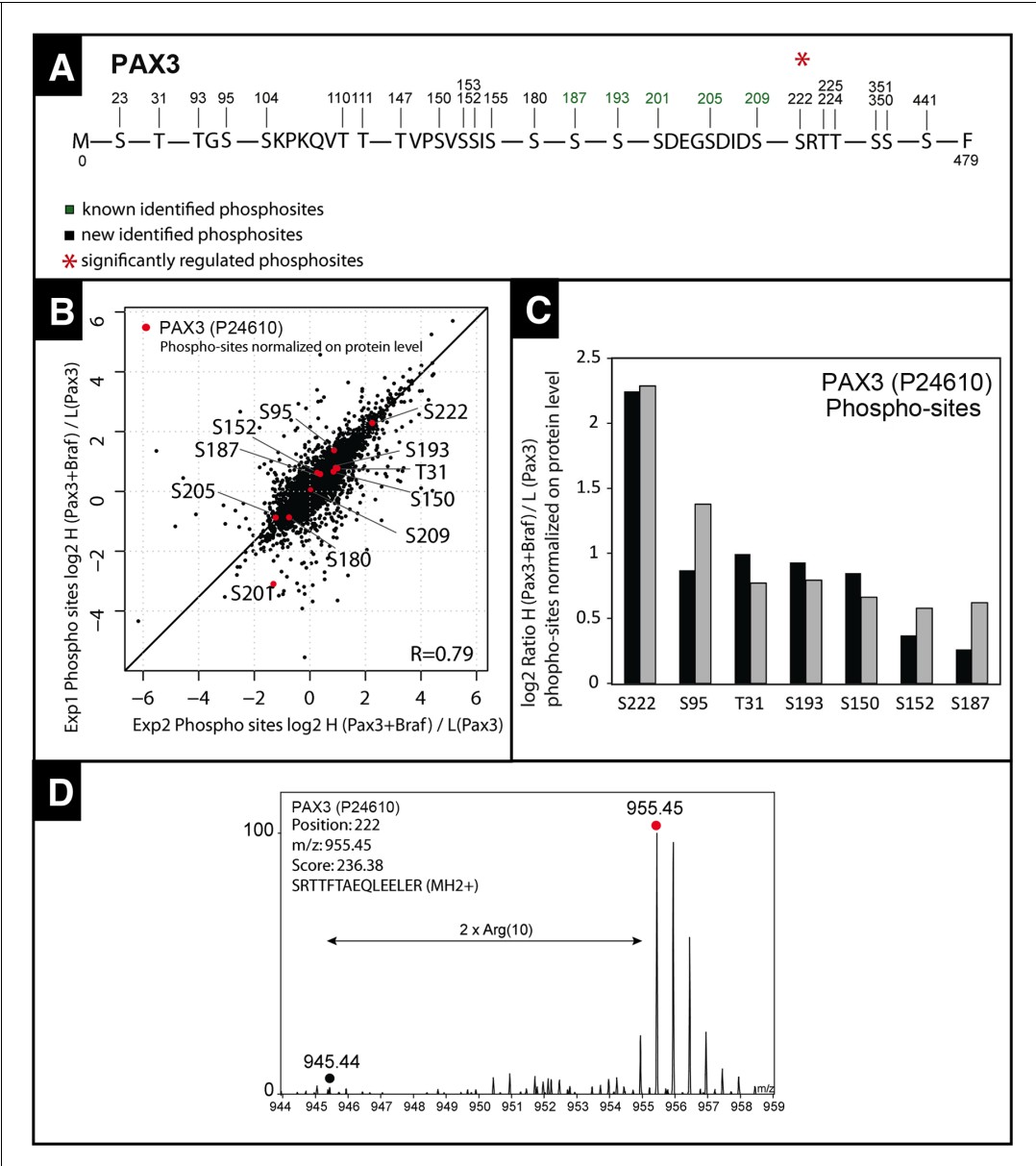

**Figure 7.** Quantitative mass spectrometry analysis of PAX3 phosphorylation sites. (**A**) Map of PAX3 phosphorylation sites determined by mass spectrometry. Sites shown in green have been published before (phosphosite.org) and sites displayed in black were newly identified. The red asterisk marks a phosphorylation, which was strongly induced by co-transfection with CA Braf (V600E). (**B**) Scatter plot showing the distribution of SILAC log2 phosphosite ratios normalized to PAX3 protein expression levels after transfection with Pax3 and Braf (heavy) and Pax3 (light). (**C**) Histogram of log2 PAX3 phosphosite ratios in two independent experiments (exp1: light grey; exp2: black) normalized to PAX3 protein expression ratios. (**D**) Representative MS SILAC spectra for identification of S222. Andromeda Score for corresponding MS/MS spectra is 236.38. The mass deviation is given in p.p.m.

The following source data is available for figure 7:

**Source data 1.** Source data for mass spectrometry analysis.

increase of serine and threonine phosphorylation by quantitative mass spectrometry after expression of CA Braf (V600E). HEK293T cells were SILAC labeled (Light: Arg0, Lys0; Heavy: Arg10, Lys8) for five passages and then transfected with Pax3 (light condition) or Pax3 plus CA Braf (V600E) (heavy condition). Protein extracts were prepared from each condition, combined at equal concentrations, processed and used for Titan (TiO$_2$) bead-based extraction of phosphorylated peptides. This approach allowed us to calculate the relative abundance of phosphorylated sites in PAX3 in response to CA BRAF (V600E) due to differential labeling with separate stable isotopes. We found that transfection of Braf primarily increased PAX3 phosphorylation at Ser222 while phosphorylation at the other sites was not prominently enhanced when normalized to PAX3 protein concentrations (*Figure 7B–D*).

The phosphorylation sites at serine residues 201, 205, 209 and 222 attracted our particular attention, since all four sites are localized within the octapeptide domain of PAX3 known to mediate several protein-protein interactions (*Miller et al., 2008*). To analyze whether any of these phosphorylation sites are instrumental for the ability of PAX3 to stimulate muscle cell migration, we replaced the respective serine residues by alanine. As expected neither WT Pax3 nor the S201A, S205A, S209A and S222A mutant constructs increased the rate of migration without activation by BRAF (*Figure 8A*) but co-transfection with CA Braf (V600E) revealed important differences. The S205A Pax3 mutant failed to stimulate muscle cell migration while S201A, S209A and S222A mutants had no discernable effects compared to WT Pax3 indicating that phosphorylation of Ser205 is instrumental for PAX3-dependent stimulation of muscle cell migration (*Figure 8A*). Since only mutation of Ser205 reduced the ability of BRAF to induce PAX3-dependent muscle cell migration, we wondered whether we had failed to detect BRAF-dependent up-regulation of PAX3 phosphorylation at this site in our quantitative mass spectrometry experiments. Therefore, we transfected Pax3 together with CA Braf (V600E) into HEK293T and C2C12 muscle cells, but this time monitored phosphorylation of PAX3 using an antibody specific for p-Ser205. We found that BRAF clearly induced phosphorylation of PAX3 at Ser205 in both cell types (*Figure 8B*). Furthermore, we performed an in vitro kinase assay using recombinant BRAF and bacterially produced, affinity-purified GST-PAX3. Western blot analysis using the p-Ser205 specific PAX3-antibody corroborated that BRAF directly phosphorylates PAX3 at Ser205 (*Figure 8C*, upper panel). To analyze the specificity of the p-Ser205 PAX3-antibody we performed in vitro kinase assays using the S201A, S205A, S209A and S222A PAX3-mutants as substrates. As expected all mutant PAX3 proteins, with the exception of S205A, reacted with the p-Ser205 PAX3 antibody after incubation with BRAF demonstrating specificity of the antibody (*Figure 8C*, lower panel). Taken together, our results suggest that BRAF phosphorylates PAX3 at multiple sites but that phosphorylation at Ser205 is critical for efficient PAX3-dependent stimulation of muscle cell migration.

## Discussion

Here, we demonstrate that BRAF is critical for migration of limb muscle precursor cells, which serves an essential role for normal limb muscle development. Targeted mutation of *Braf* in migrating limb muscle precursor cells partially phenocopied effects of inactivation of the receptor tyrosine kinase *Met* and its large adaptor protein GAB1 indicating that BRAF acts downstream of GAB1 to mediate intracellular MET signaling. Furthermore, we show that a fraction of BRAF enters the nucleus, where it binds and activates the transcription factor PAX3 thereby creating a shortcut bypassing established signaling pathways.

The crucial role of BRAF in transmitting signals downstream of MET and GAB1 does not rule out the contribution of other signaling pathways linked to GAB1 such as PI3K/AKT or PLCγ/PKC for regulation of limb muscle progenitor cell motility, proliferation and survival but firmly establishes BRAF as essential signaling hub conveying migratory signals. Interestingly, previous studies demonstrated that G-protein coupled receptor CXCR4 and GAB1 cooperate to control the development of migrating muscle progenitor cells (*Vasyutina et al., 2005*). Since CXCR4 signaling, including induction of the nuclear translocation of ERK1/2 (*Zhao et al., 2006*), relies on activation of RAS/RAF while MET associated GAB1 activates the RAS-RAF route via the tyrosine phosphatase SHP2 (*Birchmeier et al., 2003*; *Trusolino et al., 2010*), we would like to propose that important aspects of the cross-talk between the CXCR4 and MET/GAB1 signaling pathways are mediated by BRAF. However, we do not want to claim that BRAF solely exerts its effects downstream of MET. BRAF might also play an

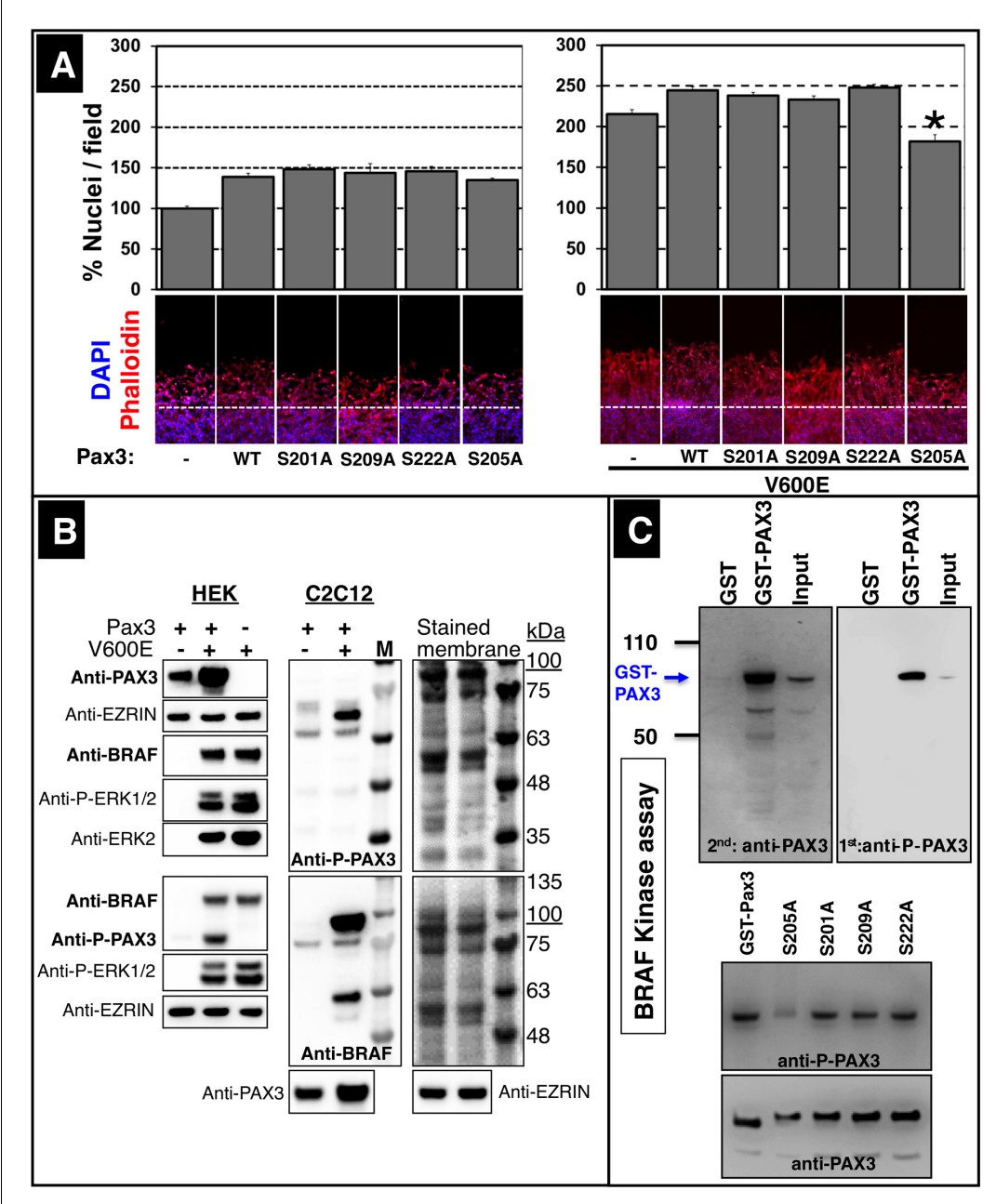

**Figure 8.** Phosphorylation of PAX3 at Ser205 is critical for stimulation of muscle cell migration. (**A**) Exchange of serine by alanine in PAX3 at Ser205 reduces C2C12 myoblast migration after transfection with CA Braf (V600E). Cultures were either transfected with a control vector (-), with wildtype Pax3 (WT) or with mutated Pax3 constructs. Mutated serine residues are indicated (S201A, S205A, S209A, S222A). Cell numbers were determined by counting the number of DAPI-stained nuclei. n = 10; Mann-Whitney-U test, (p*<0.05). (**B**) Phosphorylated PAX3 proteins were isolated from Pax3 and mutant Braf (V600E) transfected HEK293T or C2C12 cells by phospho-column affinity purification and detected by immunoblotting. Phosphorylated ERK1/2 (P-ERK1/ 2), ERK2 and EZRIN served as controls. Two sets of independent experiments are shown for HEK293T cells. Molecular sizes (in kDa) are indicated by the overlay of the colored marker with the PAX3 and BRAF bands visualized by chemiluminescence. Note that PAX3 shows a shift in the molecular weight due to the HA-tag. Stained membranes with marker are shown. (**C**) In vitro kinase assay of GST, GST-PAX3, PAX3 and different PAX3 mutants demonstrating phosphorylation of PAX3 by BRAF. Bacterially produced GST-PAX3 (WT and mutants) was purified by GST-affinity chromatography. Column eluates and inputs were incubated in vitro together with BRAF. n = 2. Phosphorylated PAX3 was detected using an antibody specific for p-Ser205 PAX3. Specificity of the p-Ser205 PAX3 antibody was assessed by the failure to detect the S205A PAX3 mutant (bottom panel).

important role in other cellular signaling pathways, which is also supported by reduced expression of *Lbx1* in *Braf* mutant embryos but not after inactivation of *Met* (*Sachs et al., 2000*). Deletion of *Braf* impairs but does not completely abolish delamination of muscle progenitors as indicated by the appearance of small numbers of committed myoblasts and the formation of residual myofibers in limbs. Hence, additional experiments are needed to determine the contribution of other signaling pathways downstream of MET and CXCR4 and to unravel the whole complexity of signaling network required for migration of limb muscle precursor cells.

Our study primarily focused on early steps of limb muscle formation ignoring potential later functions of *Braf* for myogenesis after migration is completed. Analysis of a *Met* mutant mouse strain, in which the adaptor GRB2 is unable to bind to MET, revealed that GRB2 is dispensable for migration of limb muscle precursor cells but required for proliferation of fetal myoblasts and formation of secondary myofibers (*Maina et al., 1996*). Since GRB2 is a well-known activator of RAS-RAF signaling it seems possible that loss of BRAF might also play a role at later stages of myogenesis, e.g. for generation of secondary myofibers. Moreover, PAX3, which is a target of BRAF mediated phosphorylation in the nucleus, was shown to activate the enhancer of the myogenic control factor *Myf5* in limb muscle progenitor cells (*Bajard et al., 2006*), which raises the possibility that the MET-BRAF cascade affects muscle differentiation via *Pax3*-dependent control of *Myf5* expression. Time-controlled inactivation of *Braf* during fetal myogenesis will probably resolve these issues.

Our results indicate that stimulation of myogenic cell migration and translocation of BRAF into the nucleus requires intact endosomal trafficking. Both administration of Dynasore and knockdown of *Eea1* abrogated the effects of BRAF on migration of myogenic cells and prevented accumulation of BRAF in the nucleus. Traditional models favor the view that receptors only signal at the plasma membrane and are inactivated by internalization. However, it has become clear that several receptor tyrosine kinases including EGFR, PDGFR (*Miaczynska et al., 2004a*), the insulin receptor (*Ceresa et al., 1998*) and MET (*Kermorgant et al., 2004*) signal after internalization from endosomal compartments. In fact, MET endocytic trafficking is required for the full activation of signals requiring GAB1, ERK 1/2, STAT3 and RAC1 (reviewed by (*Barrow-McGee and Kermorgant, 2014*)). Upon ligand binding, MET undergoes rapid endocytosis and traffics through peripheral endosomes to accumulate on perinuclear endosomes. MET-induced activation of STAT3 within the perinuclear compartment allows efficient activation and nuclear accumulation of STAT3 (*Kermorgant and Parker, 2008*) suggesting that trafficking and 'endosomal signaling' of MET enables direct transfer of signaling complexes to their intended target (*Miaczynska et al., 2004b*) thereby favoring distinct signaling events. Our finding that MET co-localizes with BRAF in EEA1$^+$ endosomes after HGF stimulation and that blockage of endosomal trafficking prevents accumulation of BRAF in the nucleus indicates that internalization and perinuclear accumulation of MET is instrumental for transport of a fraction of BRAF into the nucleus.

BRAF is commonly viewed as a cytoplasmic kinase and was not considered to undergo intracellular transport. Motivated by our mass spectrometry-based screening for potential interaction partners of BRAF, which resulted in the identification of the nuclear transcription factor PAX3, we used several different techniques to prove the localization of BRAF in the nucleus including subcellular fractionation and immunofluorescence using both endogenous and transfected BRAF. We reason that the low expression level of BRAF, the relatively low amount of BRAF in the nucleus and the dynamic regulation of intracellular localization might have prevented a prior detection of BRAF in the nucleus so far. In this context it is interesting to note that also other components of the signaling pathways downstream of MET undergo a cytoplasmic-nuclear transport including MKP-3 (*Karlsson et al., 2004*), ERK1/2 (*Barrow-McGee and Kermorgant, 2014*), and STAT3, which in part rely on MET-induced activation within the perinuclear compartment (*Kermorgant and Parker, 2008*) suggesting a common theme in the transfer of signaling complexes to their intended targets (*Miaczynska et al., 2004b*).

The PAX3 protein is posttranslationally modified by acetylation (*Ichi et al., 2011*), ubiquitination (*Boutet et al., 2007*), and phosphorylation (*Amstutz et al., 2008*; *Iyengar et al., 2012*; *Miller et al., 2008*). The functional relevance of PAX3 phosphorylation is largely unknown although the presence of phosphorylation sites at Ser201, Ser205 and Ser209 near the octapeptide domain, a region crucial for mediating protein-protein interactions and DNA binding suggests important regulatory functions. So far, two different kinases were identified to phosphorylate PAX3: GSK3β and CK2. CK2 seems able to phosphorylate Ser205 and Ser209 (*Iyengar et al., 2012*) while GSK3β was

found to phosphorylate Ser201, Ser205 and possibly Ser197 (*Kubic et al., 2012*). Since inhibition of GSK3β in melanoma cells leads to cellular changes paralleling PAX3 inhibition, it was claimed that GSK3β regulates proliferation and morphology of melanoma cells through phosphorylation of PAX3 (*Kubic et al., 2012*). We disclosed that BRAF phosphorylates Ser205, which is part of a cluster of serine residues (Ser187, Ser193, Ser197, Ser201, Ser205, and Ser209) required for efficient DNA binding of PAX3-FOX1A fusion proteins and necessary to institute the full transactivation potential of PAX3 in reporter gene assays (*Amstutz et al., 2008*). Serine to aspartate conversions at these sites render the transcriptional activity of PAX3/FOX1A fusion proteins insensitive to the inhibitory actions of the kinase inhibitor PKC412, which inhibits growth of alveolar rhabdomyosarcomas carrying a *Pax3/Fox1A* translocation (*Amstutz et al., 2008*). Using a different assay to monitor activity of PAX3, we found that the S205A mutation significantly inhibited PAX3-dependent C2C12 muscle cell migration while S201A, S209A and S222A mutants had no discernable effects. Based on the published observation and our own findings it seems reasonable to assume that phosphorylation of Ser205 plays an important role in regulating the biological activity of PAX3. In fact, the octapeptide domain close to Ser205 mediates interaction of PAX3 with GROUCHO, which prevents transcriptional activation of PAX3 targets during melanocyte and neural cell differentiation (*Lang et al., 2005*; *Muhr et al., 2001*). Therefore, it seems possible that phosphorylation of Ser205 or other nearby sites prevents binding of GROUCO and thereby facilitates transactivation of PAX3 target genes, although we did not investigate this possibility. Since Ser205 is phosphorylated both by CK2, GSK3β and BRAF, it is also tempting to speculate that several different signaling pathways contribute to phosphorylation of Ser205 and/or that a certain fraction of Ser205 is constitutively phosphorylated assuring a basic activity of PAX3, which is further enhanced by MET-dependent BRAF phosphorylation. However, we have to point out that we have demonstrated the critical role of Ser205 phosphorylation only in vitro in C2C12 cells and not in limb muscle progenitor cells in vivo. It is possible that different mechanisms operate in limb muscle progenitor cells than in C2C12 muscle cells, which differ in many aspects (e.g. expression of myogenic factors).

The identification of BRAF as a kinase phosphorylating PAX3 establishes a new pathway, which offers MET a direct route to regulate PAX3 activity in migrating muscle cells, independent of other intracellular phosphorylation cascades. We found that inhibition of ERK1/2 kinase did not affect BRAF-dependent stimulation of muscle precursor cell migration adding further support to the idea that BRAF exerts its activity on muscle cell migration not via the classical RAS/RAF/MEK/ERK cascade but by direct phosphorylation and activation of PAX3. On the other hand, it seems possible that RAS/RAF/MEK/ERK cascade contributes to the regulation of proliferation of fetal myoblasts and formation of secondary myofibers, which depends on MET adapter protein Grb2 (*Maina et al., 1996*). The stimulation of PAX3 activity by MET via BRAF together with the previously reported activation of the *Met* promoter by PAX3 (*Epstein et al., 1996*) suggest the existence of a positive feedback loop, which might be instrumental to maintain high levels of MET and PAX3 activity during limb muscle precursor cell migration (*Figure 9*). Importantly, we found that activation of PAX3 by overexpression of WT and CA Braf (V600E) in C2C12 muscle cells results in marked up-regulation of *Met* and *Pax3*. Similarly, WT and CA Braf (V600E) increased the transcriptional activity of PAX3 in Luciferase reporter assays on HEK293T cells, which fits to the proposed role of PAX3 to stimulate *Met* expression in myogenic cells (*Epstein et al., 1996*) and to activate its own expression by direct positive autoregulation (*Moore et al., 2013*). Moreover, activation of *Met* and *Pax3* expression by BRAF underscores the importance of BRAF in maintaining the MET-PAX3 feedback loop. It will be interesting to determine whether this loop and the signaling bypass created by BRAF-dependent PAX3 phosphorylation is also operative in other tissues and in cancer cells such as rhabdomyosarcoma and melanoma cells, which would offer new refined therapeutic approaches.

## Materials and methods

### Antibodies and reagents

C2C12 and HEK293T cells were obtained from the American Type Culture Collection (ATTC, Massnassas, Virginia) and analyzed for potential myoplasma contamination every two month using the PCR Mycoplasma Test Kit I/C from Promocell (Heidelberg, Germany). Identity of C2C12 cells was authenticated by induction of myotube differentiation. Antibodies recognizing BRAF, EEA1, LAMIN

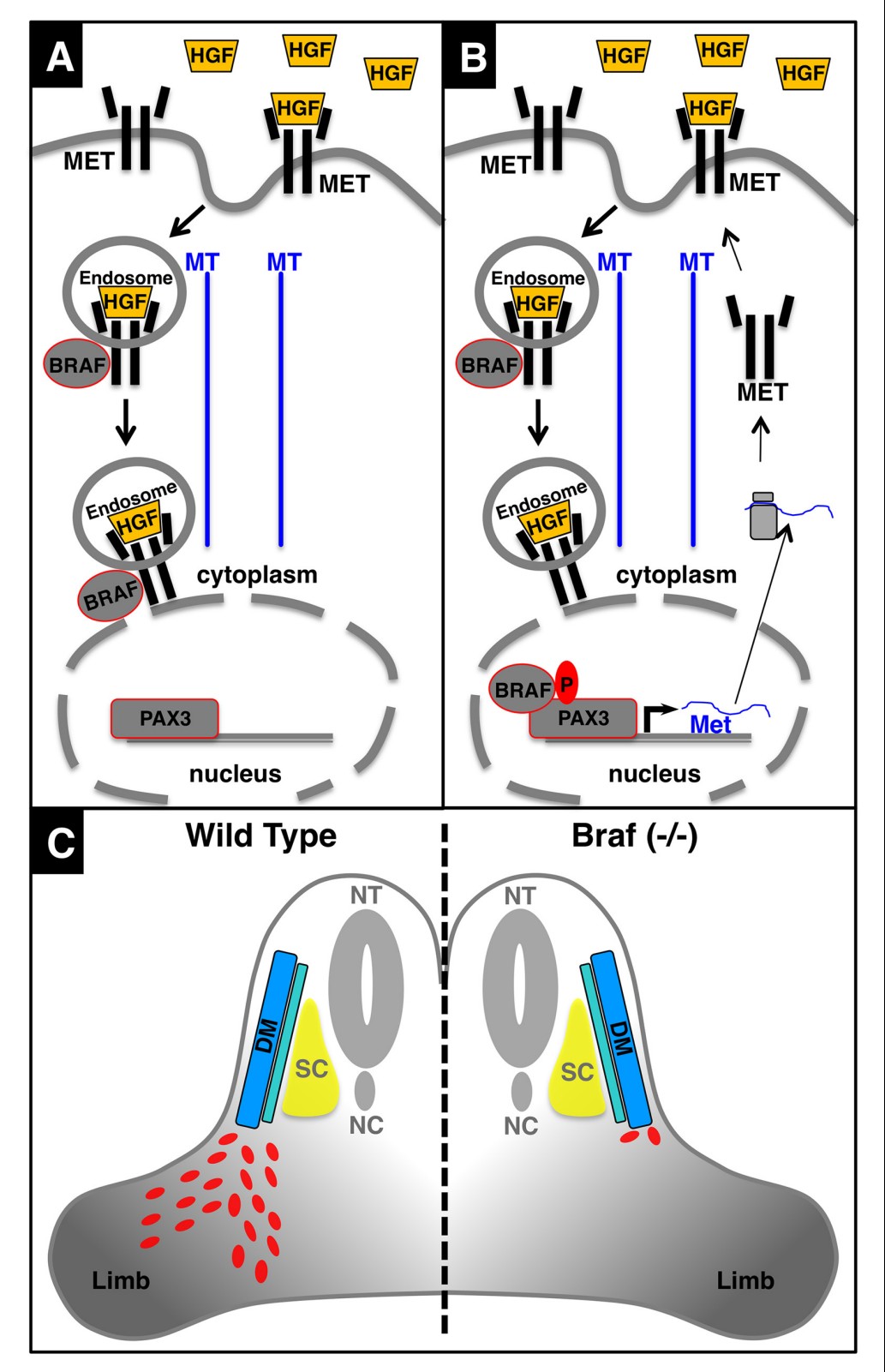

**Figure 9.** Schematic model of the MET-BRAF-PAX3 feedback loop enabling migration of limb muscle precursor cells. (**A**) Activation and subsequent internalization of MET leads to activation of BRAF and transport via endosomal trafficking to a perinuclear position. (**B**) After translocation of BRAF into the nucleus BRAF phosphorylates and activates PAX3, which promotes sustained expression of *Met* and other PAX3 target genes
*Figure 9 continued on next page*

*Figure 9 continued*

enabling limb muscle precursor cell migration. (C) Inactivation of *Braf* in PAX3-expressing cells disrupts the regulatory feedback loop leading to arrest of limb muscle precursor cell migration and defective limb muscle development. NT: neural tube; DM: dermomyotome; SC: sclerotome; NC: notochord.

A/C and MYOGENIN were obtained from BD Biosciences (San Jose, California) while PAX3, MYOD and myosin heavy chain antibodies (MHC) were from the Developmental Studies Hybridoma bank (Iowa City, Iowa). Additional PAX3 and anti-LBX1 antibodies were purchased from Abcam (Cambridge, Massachusetts). Anti-phospho-PAX3 (Serine 205) was a kind gift of Dr. A. D. Hollenbach (*Miller et al., 2008*). The antibody against CD31 and recombinant Scatter Factor/Hepatocyte Growth factor (HGF) was obtained from R and D Systems (Minneapolis, Minnesota). Anti-phospho BRAF, anti-phospho ERK1/2 and anti-GAPDH antibodies were purchased from Cell Signaling (Danvers, Massachusetts). Anti-MYF5 was from Santa Cruz (Dallas, Texas) and Anti-GST was from GE healthcare (Chicago, Illinois).

## Vectors, siRNA, pull down, kinase assay and whole mount hybridization

Plasmids containing full length wild-type (pEFm-BRAF) and mutant BRAF (pEFm-V600E) were a kind gift of Dr. R. Marais. Plasmids pcDNA-Pax3 (Addgene, Cambridge, Massachussets) and pGEX-5X-1 (Pharmacia Biotech, London, United Kingdom) were employed to construct the S205A, S201A and S222A PAX3 mutants using overlapping extension PCR as described (*Ho et al., 1989*). The luciferase reporter plasmid DNA (pGl.3 Pax3-BS luc) was constructed by the insertion of double strand oligonucleotides (5'- GCT AGC TAG CTA AAG GCA TGA CTA ATT GTA AAG GCA TGA CTA ATT GGA ATT CAG ACA CTA GAG GGT ATA TAA TGG AAG CTC GAC TTC AGG GCA AAT CCG TAA CTG TTG GTA AGC CAC CCC ATG G −3') containing a minimal promoter (from pGl.4 23, Promega, Mannheim, Germany) and two consensus sequences of PAX3 into pGl.3 (Promega) at the NheI and NcoI restriction enzyme site. Plasmid transfection of C2C12 and HEK293T cells was done with the FuGENE6 system (Roche, Mannheim, Germany) or Lipofectamin 3000 (Invitrogen, Carlsbad, California) as recommended by the manufacturers. Knock-down of different genes was achieved with the aid of siRNAs targeting Braf (SMARTpool L-040325-00-0005, NM_139294), Craf (SMARTpool L-040149-00-0005, NM_029780), Pax3 (SMARTpool L-062276-01-0005, NM_008701), Met (SMARTpool L-040878-00-0005, NM_008591), Cltc (SMARTpool L-063954-00-0005), Cav1 (SMARTpool M-058415-01-0005), Eea1 (SMARTpool L-063149-00-0005) and Dnm2 (SMARTpool L-044919-02-0005) using the corresponding transfection kit as recommended by the manufacturer (all Dharmacon, Lafayette, Colorado). Luciferase assays were done following standard procedures. GST fusion proteins expressed in E. coli were purified with Glutathione Sepharose 4B beads (Pharmacia). For pull down assays cell lysates from wild type Braf or mutant Braf transfected cultures were incubated overnight with PAX3-GST fusion proteins coupled to glutathione-Sepharose beads and processed by standard methods for Western Blot analysis. Whole-mount in situ hybridization with digoxigenin-labeled antisense cRNA probes was performed using Lbx1, Myf5, Myod, Myogenin cRNA probes as described previously (*Miller et al., 2008*; *Schäfer and Braun, 1999*). The in vitro BRAF kinase assay was performed with purified GST-proteins according to manufacturer's instructions (Millipore, Billerica, Massachusetts).

## Animals

Generation of triple *loxP*-site *Braf* mutant mice (Braf$^{nfl}$) has been described before (*Pfeiffer et al., 2013*). To remove the neomycin resistance cassette and to generate the conditional Braf$^{fl}$ allele, Braf$^{nfl}$ mice were bred with MeuCre mice (*Leneuve et al., 2003*), which show only a low cre-recombinase activity (*Leneuve et al., 2003*). Progeny carrying *loxP*-site flanked exon 3 of *Braf* (i.e. Braf$^{nfl}$ mice) were bred with Pax3-Cre mice containing a Cre-recombinase gene insertion in Exon 1 of *Pax3* (*Engleka et al., 2005*) to generate mice lacking *Braf* expression in limb muscle precursor cells. Generation of the strain is depicted in (*Figure 2*). All procedures involving animals and their care were carried out in accordance with the guidelines for animal experiments at the Max-Planck-Institute for Heart and Lung Research, which conform to the Guide for the Care and Use of Laboratory Animals (NIH Publication No. 85–23, revised 1996) and the European Parliament Directive 2010/63/EU and

the 22 September 2010 Council on the protection of animals. Animal experimentation was approved by the local Ethics committee for animal experiments at the Regierungspräsidium Darmstadt (Registration # B2/1015 „Herstellung transgener Mauslinien; Inaktivierung und Überexpression von Genen"). The animal house at the MPI-HLR is registered according to §11 German Animal Welfare Law.

## Explant cultures

For explant cultures, lateral parts of somites were dissected from HH stage 18/19 chicken embryos and cultured for one day in a collagen matrix as described previously (*Mennerich et al., 1998*). Retroviral infections of explanted tissues were performed as described (*Mennerich and Braun, 2001*). Generation of viral constructs and production of high titer virus stocks were performed as described previously (*Mennerich and Braun, 2001*). Retroviral titers ranged from $7 \times 10^7$ to $2 \times 10^9$ c.f.u./ml.

## C2C12 culture and in vitro migration assays

The C2C12 myoblast cell line was grown in DMEM (1.0 mg/ml glucose containing 10% FCS, 100 U/ml of penicillin, 100 µg/ml of streptomycin and 0.292 mg/ml L-glutamine) and used for migration assays. In the Boyden Chamber assay $2 \times 10^5$ cells were seeded into a transwell chamber of a 24-well plate, and migration was induced by 100 ng/mL HGF. After 4–16 hr cells were fixed to determine the number of migrated cells and to perform immunofluorescence analysis. Scratch assays were initiated by removing cells from a confluent (growth arrested) monolayer with a yellow tip 48 hr after transfection. After removal of non-adherent cells by washing with medium and incubation of 4 hr images were taken and six randomly selected areas of each dish were analyzed with the Image J software.

## Western blot analysis

Tissue and cell culture samples were processed for Western blot as described previously (*Poling et al., 2011*). For isolation of cytoplasmic and nuclear fractions, cells were washed three times with Hanks balanced salt solution, harvested by scraping and collected by centrifugation at 500 g at for 3 min. Different cell fractions were isolated using the NE-PER kit (ThermoFisher Scientific, Waltham, Massachusetts) following instructions of the manufacturer. For detection with the SuperSignal West Femto Maximum Sensitivity Substrate (Piercenet, Bonn, Germany) kit the secondary antibody is conjugated with horseradish peroxidase (HRP). Detection and semi-quantitative analysis of specific band patterns were performed with the Versa Doc Imaging System Model 5000 using the supporting software package Quantity One (Bio-Rad, Hercules, California).

## Immunofluorescence and in situ hybridization

Immunofluorescence staining was performed as described previously (*Kubin et al., 2011*). Apoptotic cells were detected using the TMR red in situ Cell Death Detection Kit (cat. number 12 156 792 910) from Roche) following the instructions of the manufacturer. For whole mount hybridization E10.5 to E12.5 embryos were hybridized with digoxigenin-labeled Pax3, Myod, Myf5, Lbx1 and myogenin cRNA probes. The hybridized probe was visualized using alkaline phosphatase-coupled anti-digoxigenin antibody and nitroblue tetrazolium-5-bromo-4-chloro-3-indolylphosphate substrate according to manufacturer's recommendations (Boehringer Mannheim, Mannheim, Germany).

## Immunoprecipitation and in-gel digestion

For immunoprecipitation E10.5 C57Bl6/J limb buds endogenously expressing BRAF and C2C12 cells transfected with mutant Braf (pEFm-V600E) were used. Cells were lysed in ice-cold radioimmunoprecipitation (RIPA) buffer (100 mM Tris pH 7.5, 300 mM NaCl, 2% NP-40, 2 mM EDTA, 0.2% sodium deoxycholate and 1:100 protease inhibitor cOmplete (from Roche)). Protein extracts were purified by centrifugation and incubated with the antibody against endogenous BRAF in presence of Sepharose G beads (Life Technologies, Carlsbad, California). Precipitated proteins were then centrifuged and washed several times using ice-cold RIPA buffer. Finally, antibody–antigen complexes were eluted by boiling in LDS buffer (Novex by Life Technologies) and samples were subjected to in-gel digestion. Immunoprecipitated proteins were separated according to their molecular weight by subjecting them to SDS-PAGE (4–12% NuPage BisTris Gel, Invitrogen) followed by Colloidal Blue

staining (Expedeon, San Diego, California). Gel lanes were cut into equal pieces and digested in the gel as described by Shevchenko et al. (*Shevchenko et al., 2006*). In brief, gel pieces were washed, de-stained and dehydrated. Proteins were reduced with 10 mM dithiothreitol (DTT), alkylated with 55 mM iodoacetamide (IAA) and digested with the endopeptidase sequencing-grade Trypsin (Promega) overnight. Generated peptides were extracted using an increasing acetonitrile concentration. Collected peptide mixtures were concentrated and desalted using the Stop and Go Extraction (STAGE) technique (*Rappsilber et al., 2003*).

## Protein digestion and phosphopeptide enrichment

HEK293T cells were SILAC labeled (Light: Arg0, Lys0; Heavy: Arg10, Lys8) for at least five passages and then transfected with pcDNA-Pax3 (light condition) or pcDNA-Pax3 plus mutant Braf (pEFm-V600E in heavy condition). After 36 hr of transfection, the cells were homogenized in SDS lysis buffer (4% SDS/0.1 M Tris/HCl pH 7.6) followed by sonication and boiling at 70°C for 10 min. Lysates were clarified by centrifugation (13,000g, 10 min) and protein concentration was determined using DC assay (Bio-Rad). Approximately 5 mg protein of each labeling condition were pooled and reduced using 100 mM dithiothreitol (Sigma-Aldrich, Taufkirchen, Germany) for 10 min at 56°C and then subjected to the FASP digestion technique (*Wiśniewski et al., 2009*). Briefly, samples were washed with 8 M urea, alkylated with 55 mM iodoacetamide (Sigma-Aldrich) and digested overnight in 20 mM ammoniumbicarbonate/trypsin (Promega), at an enzyme-to-protein ratio of 1:100. For fractionation by SCX, the FASP-digested peptides were collected by multiple washing of filter units, acidified to pH 2.67 with trifluoroacetic acid (TFA) and filled up with ACN to a final concentration of 30%. The samples were loaded on a ResourceS 1 ml SCX column (Äkta Purifier, GE Healthcare). Flow-through was collected and peptides were separated according to their charge in acidic conditions using a linear increase in salt concentration in a binary buffer system: buffer A 7 mM $KH_2PO_4$ in 30% acetonitrile (ACN) (pH 2.65) and B 7 mM $KH_2PO_4$, 350 mM KCl in 30% ACN (pH 2.65). All fractions were pooled conducting absorbance at 280 nm to a total of 8–10 fractions, concentrated and adjusted to binding conditions for Titan sphere ($TiO_2$) bead-based extraction of phosphorylated peptides (93% acetonitrile, 7% TFA). Fractions were incubated twice with 2.5 mg of $TiO_2$ beads and flow-throughs were incubated three times with 5 mg of $TiO_2$ beads (SLSC Science, St. Louis, Missouri). Beads were washed several times with decreasing content of TFA (6–3%) and loaded on C8 material-containing tips. Peptides were eluted with 40% ammonia/acetonitrile (pH 11.6), concentrated in a speedvac at room temperature to almost complete dryness and diluted in acidified (0.1% formic acid or 0.5% acetic acid) $H_2O$ before mass spectrometry. The experiment was performed in duplicates.

## Liquid chromatography and mass spectrometry

Instrumentation for LC-MS/MS analysis consisted of a NanoLC 1000 coupled via a nano-electrospray ionization source to the quadrupole-based Q Exactive benchtop mass spectrometer (*Michalski et al., 2011*). Peptide separation was carried out according to their hydrophobicity on an in-house packed 50 cm column with 1.9 mm C18 beads (Dr Maisch GmbH, Ammerbuch, Germany) using a binary buffer system consisting of solution A: 0.1% formic acid (0.5% acetic acid) and B: 80% acetonitrile, 0.1% formic acid (80% acetonitrile, 0.5% acetic acid). Linear gradients from 7–38% B in 150 were applied with a following increase to 80% B within 5 min and a re-equilibration to 5% B (*Kruger et al., 2008*). MS spectra were acquired using 1E6 as an AGC target, a maximal injection time of 20 ms and a 70,000 resolution at 200 m/z. A Top10 method was applied for subsequent acquisition of higher-energy collisional dissociation (HCD) fragmentation MS/MS spectra of the top 10 most intense peaks. Resolution for MS/MS spectra was set to 35,000 at 200 m/z, AGC target to 5E5, max injection time to 120 ms and the isolation window to 1.7 Th.

## Mass spectrometry data analysis

All acquired raw files were processed using MaxQuant (1.5.3.12) (*Cox and Mann, 2008*) and the implemented Andromeda search engine (*Cox et al., 2011*). For protein assignment, spectra were correlated with the Uniprot human database (v. 2015) including a list of common contaminants. Searches were performed with tryptic specifications and default settings for mass tolerances for MS and MS/MS spectra. Carbamidomethyl at cysteine residues was set as a fixed modification, while oxidations at methionine, acetylation at the N-terminus were defined as variable modifications. The

minimal peptide length was set to seven amino acids, and the false discovery rate for proteins and peptide-spectrum matches to 1%. The match-between-run feature with a time window of 1 min was used. For further analysis, the Perseus software (1.5.0.31) was used and first filtered for contaminants and reverse entries as well as proteins that were only identified by a modified peptide. The SILAC Ratios were logarithmized and grouped into duplicates and a one-sample t-test was performed. Probability values (p)<0.05 were considered statistically significant.

## Statistical analysis

Statistical analysis was performed with the Graph Pad Prism 5.0a (GraphPad Software, La Jolla, California) using the Mann-Whitney-U test or ANOVA with a Tukey-Kramer post-hoc comparison test as indicated. Probability values (p)<0.05 were considered statistically significant. The n-numbers always represent biological replicates and are indicated in the figure legends together with the p-values. Mean and SD or SEM values are indicated within the figures.

## Acknowledgements

The authors thank Katja Kolditz, Kerstin Richter, Jutta Wetzel and Brigitte Matzke for technical assistance. The authors are indebted to Prof. Ulf Rapp for the generous gift of Braf triple *loxP* mice and to Dr. Andrew D. Hollenbach for providing Anti-phospho-PAX3 (Serine 205) antibodies.

## Additional information

### Funding

| Funder | Grant reference number | Author |
|---|---|---|
| Deutsche Forschungsgemeinschaft | SFB TRR81 TP02 | Thomas Braun |
| LOEWE Center for Cell and Gene Therapy | | Thomas Braun |
| German Center for Cardiovascular Research | | Thomas Braun |
| Fondation Leducq | MolCellCard | Thomas Braun |

The funders had no role in study design, data collection and interpretation, or the decision to submit the work for publication.

### Author contributions

JS, TK, TB, Conception and design, Acquisition of data, Analysis and interpretation of data, Drafting or revising the article; SW, SH, MK, Acquisition of data, Analysis and interpretation of data, Drafting or revising the article; SK, Acquisition of data, Analysis and interpretation of data; JP, Conception and design, Drafting or revising the article

### Author ORCIDs

Sawa Kostin, http://orcid.org/0000-0002-1594-9476
Thomas Braun, http://orcid.org/0000-0002-6165-4804

### Ethics

Animal experimentation: All procedures involving animals and their care were carried out in accordance with the guidelines for animal experiments at the Max-Planck-Institute for Heart and Lung Research, which conform to the Guide for the Care and Use of Laboratory Animals (NIH Publication No. 85-23, revised 1996) and the European Parliament Directive 2010/63/EU and the 22 September 2010 Council on the protection of animals. Animal experimentation was approved by the local Ethics committee for animal experiments at the Regierungspräsidium Darmstadt (Registration # B2/1015 „Herstellung transgener Mauslinien; Inaktivierung und Überexpression von Genen"). The animal house at the MPI-HLR is registered according to §11 German Animal Welfare Law.

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
