## [Decision Letter]

Thank you for submitting your article "B-Raf activates Pax3 to control muscle precursor cell migration during forelimb muscle development" for consideration by *eLife*. Your article has been reviewed by three peer reviewers, and the evaluation has been overseen by a Reviewing Editor and Fiona Watt as the Senior Editor. The following individuals involved in review of your submission has agreed to reveal his identity: Frederic Relaix (Reviewer #2).

The reviewers have discussed the reviews with one another and the Reviewing Editor has drafted this decision to help you prepare a revised submission.

This paper provides new insight into the molecular mechanisms that underlie the Pax3 dependent regulatory cascade that leads to the migration of myogenic progenitor cells from the dermomyotome of the somite into the limb. The authors show that BRaf interacts with Pax3 leading to Pax3 phosphorylation which is required for myogenic cell migration and limb muscle formation. The conclusions are based on careful work using sophisticated technologies. However, before acceptance the following points should be addressed:

Major point requiring presentation of further data:

The phenotype of the conditional BRaf mutant should include documentation of the state of the hypaxial dermomyotome at limb level at E9.5 and E10.5 (Figure 3). The deficit in limb muscle formation may be due to failure to migrate or to a change in progenitor cell survival/proliferation. Please provide better resolution pictures of the dermomyotomal lip in the mutant, and provide data on the absence of apoptosis. Do the progenitors remain in the dermomyotome? The expression of BRaf in the dermomyotome should be documented.

Other points:

1) Figure 1 requires clarification. Figure 1. Does CA BRaf affect proliferation or migration? In Figure 1, (also 6C) what do the numbers on the vertical axis refer to? What is the meaning of the different type of fillings in the bars? Figure 1, please explain what these inhibitors do in the figure legends. On the graph in 1C, there is a ** significant only for the Dynasore + V600E. This would indicate that the reduction of migration is only significant when the CA BRaf is used (V600E). This is not clear in the text, or in the figure legend.

2) In Figure 5, co-localization of Pax3 and BRaf in nuclei in vivo should be shown with a wider field to include more than a single Pax3+ nucleus.

3) "Statistical analysis was performed with the Graph Pad Prism software using the t-test". The T-test is used for normal populations. In cases where the population is not normal (which is most likely the case, given the low n) non-parametric statistical analysis should be used. More information on the method of cell counting underlying the graphs in A would be useful. Was this based on counting cell nuclei using DAPI? On the graphs, are the numbers on the left indicating cell number? This should be indicated.

4) Pax3 is a poor transcription factor in transactivation studies on target promoters or polymerised Pax3 binding sites. It would be interesting to see the effect of CA B-Raf (V600E) on Pax3 target activation in vitro, for example on c-Met regulatory elements. If the authors perform such experiments, they should be careful to use the PAX3 binding sites described by Cao et al. in the PAX3-FOXO1A ChIP-seq study.

5) A paper from Yvergoneau et al. (2012) (see also Mayeuf et al., 2016) has shown that when VEGFR2 is knocked out in paraxial mesoderm, muscle progenitors from the lateral dermomyotome do not migrate into the limb bud. This paper thus assigned a seemingly crucial role to endothelial precursors in the subsequent migration of muscle progenitors into the limb. VEGFR2 signals among other things through Raf and the endothelial precursors present in the lateral dermomyotome would certainly be affected in the Pax3-Cre/B-Raf del/del. Can the authors rule out that the effect they observe here is not due to a problem of signaling downstream of Flk1? Obviously, the "inhibition of HGF-mediated migration of myogenic cells by knockdown of Pax3" (subsection “B-Raf regulates muscle precursor cell migration through Pax3 phosphorylation”) is not really a strong argument, since Pax3 is probably upstream of VEGFR2 as well.

6) One should be very cautious about over-expression studies. Although the subcellular fractionation experiments seem to indicate that BRaf goes into the nucleus, it would be important to show data on the purity of these fractions, (the subcellular fractionation technique does not appear to be in the Materials and methods section). The authors indicate that the Met receptor traffics from the plasma membrane to perinuclear endosomes. Is it possible that the IF staining observed with BRaf lights up perinuclear endosomes? Could the authors either nail down this possibility or be more careful on the wording.

7) The finding that B-Raf phosphorylates Pax3 at sites located in the octapeptide sequence, a domain previously described as mediating interaction with Groucho repressor proteins suggests a possible interplay between repression and activation. This could be discussed.

8) In a number of figures, please provide data on the efficiency of siRNAs.

9) In Met mutants, Lbx1 is induced, indicating that the effects the authors observe in vivo are not solely downstream effects of Met. Please discuss.

---

## [Author Response]

*This paper provides new insight into the molecular mechanisms that underlie the Pax3 dependent regulatory cascade that leads to the migration of myogenic progenitor cells from the dermomyotome of the somite into the limb. The authors show that BRaf interacts with Pax3 leading to Pax3 phosphorylation which is required for myogenic cell migration and limb muscle formation. The conclusions are based on careful work using sophisticated technologies. However, before acceptance the following points should be addressed:*

*Major point requiring presentation of further data:*

*The phenotype of the conditional BRaf mutant should include documentation of the state of the hypaxial dermomyotome at limb level at E9.5 and E10.5 (Figure 3). The deficit in limb muscle formation may be due to failure to migrate or to a change in progenitor cell survival/proliferation. Please provide better resolution pictures of the dermomyotomal lip in the mutant, and provide data on the absence of apoptosis. Do the progenitors remain in the dermomyotome? The expression of BRaf in the dermomyotome should be documented.*

We thank the reviewers for this important comment. We performed additional immunofluorescence analysis of B-Raf mutant embryos at E10.5 using antibodies directed against Pax3, which labels cells in the dermomyotome and migrating muscle progenitor cells. We found that inactivation of B-Raf in Pax3-expressing cells resulted in a dramatic reduction of the number of delaminated Pax3-positive cells migrating towards the muscle anlagen in the limb. The delamination of muscle progenitor cells was not completely blocked as indicated by the presence of few Pax3-positive cells outside of the dermomyotomal epithelium, which corresponds nicely to the reduction of limb muscles in B-Raf mutant embryos.

Furthermore, we conducted a TUNEL staining of WT and B-Raf mutant embryos at E10.5, which did not reveal a significant increase of apoptotic cells suggesting that inactivation of B-Raf did not lead to a major increase of programmed cells death. In addition, we investigated the distribution of B-Raf protein in WT and B-Raf mutant embryos at E10.5. We found that B-Raf is strongly expressed in the dermomyotome of E10.5 WT embryos. As expected, we did not find B-Raf expression in somites of Pax3-Cre//B-Raf mutants.

*Other points:*

*1) Figure 1 requires clarification. Figure 1. Does CA BRaf affect proliferation or migration?*

The extent of cell migration in Figure 1 was determined 4 hours after scratching of the quiescent confluent (growth arrested) cell monolayer. Since re-initiation of cell proliferation occurs several hours later (after 12-16 hours), we concluded that the increase of cells in the scratched area is due to migration and not due to proliferation. We have now clearly indicated the time point when the samples were taken in the legend to Figure 1.

*In Figure 1, (also 6C) what do the numbers on the vertical axis refer to?*

The number on the vertical axis in Figure 1 and Figure 6 refers to counted nuclei in the scratched fields. Since C2C12 cells are mononuclear prior to fusion, the nuclei count reflects the number of cells. We have now properly labeled the vertical axis in the revised manuscript.

*What is the meaning of the different type of fillings in the bars?*

The different types of fillings in the bars have no specific meaning and were only meant to make to figure more accessible. To avoid confusion, we have now used the same filling for all bars.

*Figure 1, please explain what these inhibitors do in the figure legends.*

CytoD (cytochalasin D) is a cell permeable toxin that binds to the barbed end of actin filaments and inhibits both association and dissociation of their subunits thus causing inhibition of actin polymerization and disruption of actin filaments. Noco (Nocodazole) is an agent that reversibly interferes with the polymerization of microtubules. U0126 is a small molecule that specifically inhibits the MEK/ERK pathway. Dyna (Dynasore) is a GTPase inhibitor that targets dynamin-1, dynamin-2 as well as Drp1 (mitochondrial) and blocks dynamin-dependent endocytosis. Colch (colchicine) is an alkaloid that binds to tubulin preventing microtubule polymerization. Colchicine inhibits mitosis and endocytosis. MßCD (methyl-β-cyclodextrin) removes cholesterol from cultured and disrupts lipid rafts thus blocks formation of endocytic vesicles. Paclit (paclitaxel) binds to tubulin and disrupts the disassembly of microtubuli thereby preventing mitosis and endocytosis. We have added this information to the legend of Figure 1 as requested.

*On the graph in 1C, there is a ** significant only for the Dynasore + V600E. This would indicate that the reduction of migration is only significant when the CA BRaf is used (V600E). This is not clear in the text, or in the figure legend.*

The reviewers are right. Significance is only achieved when the activated version of B-Raf is used. We now mention this finding in the text. The reason for this observation is the much stronger ability of CA B-Raf to stimulate migration compared to WT B-Raf.

*2) In Figure 5, co-localization of Pax3 and BRaf in nuclei* in vivo *should be shown with a wider field to include more than a single Pax3+ nucleus.*

We have replaced panel D in Figure 4, which now provides a wider field with several cells Pax3-positive nuclei containing B-Raf.

*3) "Statistical analysis was performed with the Graph Pad Prism software using the t-test". The T-test is used for normal populations. In cases where the population is not normal (which is most likely the case, given the low n) non-parametric statistical analysis should be used.*

We assumed a normal distribution of measurements, which is common practice. In fact, I have rarely seen manuscripts in which normal distributions of measurements were calculated, although the authors used a parametric t-test. However, the reviewers are of course right that such an assumption cannot be taken for granted. We have therefore repeated the calculations using the non-parametric Mann-Whitney U test leading to essentially the same conclusions, although the significance level was lower in some experiments when compared to parametric t-tests. The corresponding figures were changed accordingly.

*More information on the method of cell counting underlying the graphs in A would be useful. Was this based on counting cell nuclei using DAPI? On the graphs, are the numbers on the left indicating cell number? This should be indicated.*

C2C12 cells contain single nuclei prior to fusion. We therefore determined the number of cells by counting the number of DAPI-positive nuclei. The numbers at the graphs indicate the relative number of nuclei (or cells), which has been now clearly indicated.

*4) Pax3 is a poor transcription factor in transactivation studies on target promoters or polymerised Pax3 binding sites. It would be interesting to see the effect of CA B-Raf (V600E) on Pax3 target activation* in vitro*, for example on c-Met regulatory elements. If the authors perform such experiments, they should be careful to use the PAX3 binding sites described by Cao et al. in the PAX3-FOXO1A ChIP-seq study.*

The reviewers raised an interesting point, which we were happy to address. We generated a luciferase reporter construct containing two consensus Pax3 binding sites in front of a minimal promoter and performed co-transfection experiments together with a Pax3 expression construct either in the presence of absence of constitutively active B-Raf. Importantly, we observed a dramatic increase in reporter gene activity when constitutively active B-Raf was co-transfected. Co-transfection of WT B-Raf did also stimulate transcription of the reporter construct but to a lesser extent than the constitutively active B-Raf version.

*5) A paper from Yvergoneau et al. (2012) (see also Mayeuf et al., 2016) has shown that when VEGFR2 is knocked out in paraxial mesoderm, muscle progenitors from the lateral dermomyotome do not migrate into the limb bud. This paper thus assigned a seemingly crucial role to endothelial precursors in the subsequent migration of muscle progenitors into the limb. VEGFR2 signals among other things through Raf and the endothelial precursors present in the lateral dermomyotome would certainly be affected in the Pax3-Cre/B-Raf del/del. Can the authors rule out that the effect they observe here is not due to a problem of signaling downstream of Flk1? Obviously, the "inhibition of HGF-mediated migration of myogenic cells by knockdown of Pax3" (subsection “B-Raf regulates muscle precursor cell migration through Pax3 phosphorylation”) is not really a strong argument, since Pax3 is probably upstream of VEGFR2 as well.*

Our analysis focused on the migration of muscle progenitor cells downstream of Pax3 and B-Raf and not on the migration of endothelial cells. Furthermore, VEGFR2 (commonly known as Flk1 or KDR) is assumed to signal via the PKC-MAPK and the PI3K pathway although some evidence exists that Flk1 indeed might also activate MAPK via C-Raf and MEK in some cells types. Hence, it does not seem very likely that B-Raf plays a major role downstream of Flk1. Nevertheless, we investigated the possibility that indirect effects of B-Raf in endothelial precursor cells downstream of Flk1 might be important for migration of muscle progenitors into the limb. To approach this problem, we performed immunostaining of WT and B-Raf mutant embryos at E10.5 using antibodies against CD31, which identifies endothelial cells in the developing embryo. Similar to the situation in Pax3 mutant embryos, we did not observe a significant reduction of endothelial cells in B-Raf mutant limbs suggesting that B-Raf does not play a crucial role downstream of Flk1 for migration of endothelial precursor cells.

*6) One should be very cautious about over-expression studies. Although the subcellular fractionation experiments seem to indicate that BRaf goes into the nucleus, it would be important to show data on the purity of these fractions, (the subcellular fractionation technique does not appear to be in the Materials and methods section). The authors indicate that the Met receptor traffics from the plasma membrane to perinuclear endosomes. Is it possible that the IF staining observed with BRaf lights up perinuclear endosomes? Could the authors either nail down this possibility or be more careful on the wording.*

Of course, the reviewers are right that overexpression studies need to be viewed with caution and are a source of potential artifacts. We have also added an additional sentence to emphasize the limitations of overexpression studies. However, we would like to emphasize that in addition to the overexpression studies we performed co-IPs of endogenous Pax3 and B-Raf and demonstrated the presences of B-Raf in nuclei using sections of embryonic fore limbs. The reviewers might have noticed that we controlled the purity of subcellular fractions by probing for GAPDH (a cytosolic protein) and lamin a/c (nuclear proteins). As shown in Figure 5 the nuclear fractions do not even contain traces of GAPDH indicating that the nuclear fractions are indeed “clean”. We are sorry that we did not describe the method used for subcellular fractionation in the Methods section. We have used the NE-PER kit for cultured cells from Thermo Fischer Scientific for this purpose, which is now described in the Methods section.

Moreover, we have performed additional confocal microscopy using antibodies directed against B-Raf and the endosomal marker EEA1, which demonstrates that the B-Raf signal in the nucleus is not associated with the endosomal marker (new panel in (Figure 5—figure supplement 1). The distribution of endosomes, as visualized by staining for EEA1, partially overlaps with the perinuclear accumulation of B-Raf but not with the presence of B-Raf within the nucleus. In addition, we would like to point out that the confocal z-stacks of B-Raf (Figure 5—figure supplement 1) clearly indicate the presence of B-Raf WITHIN the nucleus. Taken together, we believe that we did everything we could to demonstrate the presence of B-Raf within nuclei.

*7) The finding that B-Raf phosphorylates Pax3 at sites located in the octapeptide sequence, a domain previously described as mediating interaction with Groucho repressor proteins suggests a possible interplay between repression and activation. This could be discussed.*

We thank the reviewer for this hint. Of course, it is possible that phosphorylation of Pax3 by B-Raf in the octapeptide domain results in decreased binding of Groucho repressor proteins thereby leading to increased Pax3 activity. However, we did not investigate this possibility, which needs careful and detailed studies that seem outside of the scope of the current manuscript. Nevertheless, we have added a short paragraph to discuss such a scenario.

8) In a number of figures, please provide data on the efficiency of siRNAs.

We apologize for the omission. Of course, we determined the efficiency of siRNA knockdowns for individual target RNAs. We have now state the knockdown efficiencies in the figure legends.

*9) In Met mutants, Lbx1 is induced, indicating that the effects the authors observe* in vivo *are not solely downstream effects of Met. Please discuss.*

We thank the reviewers for this hint. We do not claim that B-Raf exerts its effects *solely* by mediating downstream effects of the c-met receptor. It is certainly possible that the downregulation of Lbx1 in B-Raf mutants is due to reduced activity of other pathways, which might explain why Lbx1 expression is reduced in B-Raf but not in c-met mutants as shown e.g. in the paper from (Sachs et al. JCB 150, 1375-1384, 2000). We now discuss this issue in the revised manuscript.